# Macrophage ATP citrate lyase deficiency stabilizes atherosclerotic plaques

Jeroen Baardman [1,12], Sanne G. S. Verberk [2,12], Saskia van der Velden[1], Marion J. J. Gijbels[1,3], Cindy P. P. A. van Roomen[1], Judith C. Sluimer [3,4], Jelle Y. Broos[5,6], Guillermo R. Griffith [1], Koen H. M. Prange[1], Michel van Weeghel [7,8], Soufyan Lakbir[2,9], Douwe Molenaar[9], Elisa Meinster[2], Annette E. Neele[1], Gijs Kooij[5], Helga E. de Vries[5], Esther Lutgens [1,10], Kathryn E. Wellen [11], Menno P. J. de Winther [1,10✉] & Jan Van den Bossche [1,2✉]

Macrophages represent a major immune cell population in atherosclerotic plaques and play central role in the progression of this lipid-driven chronic inflammatory disease. Targeting immunometabolism is proposed as a strategy to revert aberrant macrophage activation to improve disease outcome. Here, we show ATP citrate lyase (Acly) to be activated in inflammatory macrophages and human atherosclerotic plaques. We demonstrate that myeloid Acly deficiency induces a stable plaque phenotype characterized by increased collagen deposition and fibrous cap thickness, along with a smaller necrotic core. In-depth functional, lipidomic, and transcriptional characterization indicate deregulated fatty acid and cholesterol biosynthesis and reduced liver X receptor activation within the macrophages in vitro. This results in macrophages that are more prone to undergo apoptosis, whilst maintaining their capacity to phagocytose apoptotic cells. Together, our results indicate that targeting macrophage metabolism improves atherosclerosis outcome and we reveal Acly as a promising therapeutic target to stabilize atherosclerotic plaques.

[1] Department of Medical Biochemistry, Experimental Vascular Biology, Amsterdam Infection and Immunity, Amsterdam Cardiovascular Sciences, Amsterdam UMC, University of Amsterdam, Amsterdam, Netherlands. [2] Department of Molecular Cell Biology and Immunology, Amsterdam Cardiovascular Sciences, Cancer Center Amsterdam, Amsterdam UMC, Vrije Universiteit Amsterdam, Amsterdam, Netherlands. [3] Department of Pathology and Molecular Genetics, CARIM, Maastricht University, Maastricht, Netherlands. [4] BHF Centre for Cardiovascular Sciences (CVS), University of Edinburgh, Edinburgh, UK. [5] Department of Molecular Cell Biology and Immunology, Amsterdam Neuroscience, MS Center Amsterdam, Amsterdam UMC, Vrije Universiteit Amsterdam, Amsterdam, Netherlands. [6] Leiden University Medical Center, Center for Proteomics & Metabolomics, Leiden, Netherlands. [7] Laboratory Genetic Metabolic Diseases, Amsterdam Cardiovascular sciences, Amsterdam UMC, University of Amsterdam, Amsterdam, Netherlands. [8] Core Facility Metabolomics, Amsterdam UMC, University of Amsterdam, Amsterdam, Netherlands. [9] Systems Bioinformatics, Vrije Universiteit Amsterdam, Amsterdam, Netherlands. [10] Institute for Cardiovascular Prevention (IPEK), Ludwig Maximilians University, Munich, Germany. [11] Department of Cancer Biology, Abramson Family Cancer Research Institute, Perelman School of Medicine, University of Pennsylvania, Philadelphia, PA, USA. [12] These authors contributed equally: Jeroen Baardman, Sanne G. S. Verberk. ✉email: m.dewinther@amsterdamumc.nl; j.vandenbossche@amsterdamumc.nl

Atherosclerosis is a lipid-driven chronic inflammatory disorder of the arteries in which macrophages are the most prevalent immune cells and define disease development[1,2]. One of the crucial initiating events in atherosclerosis is the infiltration and accumulation of cholesterol-rich lipoproteins within the intima of the arterial vessel wall. Subsequent modification of such lipoproteins induces an inflammatory response and as a result, monocytes are recruited to the vessel wall and differentiate into macrophages. Although macrophages can worsen disease progression by propagating inflammation, they can also stabilize atherosclerotic plaques by promoting the formation of a fibrous cap and by clearing apoptotic cells (ACs) to prevent necrotic core formation. Uptake of cholesterol gives macrophages their foamy appearance and this can lead to apoptosis[3]. Initially, ACs are efficiently cleared by neighboring macrophages in a protective process termed efferocytosis[4]. In advanced atherosclerosis, insufficient efferocytosis induces AC accumulation and necrotic core formation. In combination with thinning of the protective fibrous cap this can make plaques vulnerable to rupture and can cause cardiovascular events such as myocardial infarction and stroke[5].

Given the prominent role of macrophages in the progression of atherosclerosis, modulating their responses is considered as a promising strategy to treat atherosclerosis[6]. The last decade, modulation of intracellular metabolic pathways has emerged as a new tool to reshape deranged macrophage functions and is considered as a new therapeutic opportunity[6–9]. Indeed, recent immunometabolism research highlights that intracellular metabolic reprogramming is a key controller of macrophage activation[7]. High rates of glycolysis supports the inflammatory properties op macrophages that are activated with lipopolysaccharide (LPS), whereas increased mitochondrial oxidative phosphorylation supports interleukin (IL)-4-induced macrophage responses. Recent literature revealed an important role for ATP citrate lyase (Acly) in translating metabolic changes into altered macrophage phenotype and function[10–12].

Acly is a key metabolic enzyme that converts mitochondria-derived citrate into acetyl-CoA and oxaloacetate within the cytosol. Acly-dependent acetyl-CoA incorporation into histone promotes chromosome accessibility and regulates both LPS- and IL-4-induced macrophage activation[10,11]. In addition, acetyl-CoA fuels the synthesis of fatty acids, cholesterol, and the acetylation of non-histone proteins[13,14]. Serving as a link between carbohydrate and lipid metabolism makes Acly a promising target to lower low-density lipoprotein (LDL) cholesterol levels and to reduce cardiovascular risk[15–17]. Bempedoic acid is a competitive Acly inhibitor that is specifically activated in hepatocytes where it upregulates LDL receptor (LDLr) expression. Although promising data from clinical trials highlight that hepatic Acly inhibition reduces LDL cholesterol levels in hypercholesterolemic patients, the tools to study this key metabolic hub in macrophages in vivo remained absent.

Here, we first demonstrate that Acly is activated in inflammatory macrophages and in human atherosclerotic plaques. Next, we define its formerly unknown role in regulating both plaque and macrophage phenotype. Using a conditional genetic knockout mouse model, we find that Acly deficiency in myeloid cells induces a stable plaque phenotype as demonstrated by increased collagen content and fibrous cap thickness, along with a decreased necrotic core size. Further functional, lipidomic, and transcriptional characterization in vitro show that this in vivo plaque phenotype is linked to deregulated fatty acid and cholesterol biosynthesis and reduced liver X receptor (LXR) activation within the macrophages. This results in macrophage apoptosis, along with more efficient efferocytosis and increased clearance ACs. As such, we show that targeting macrophage Acly improve

atherosclerosis outcome and can serve as a promising therapeutic target to stabilize atherosclerotic plaques.

## Results

**Acly is activated in inflammatory conditions.** Acly can be regulated at distinct levels, with phosphorylation at serine 455 serving as a key posttranslational modification to promote its enzymatic activity[18]. To study Acly expression and activity in inflammatory conditions, we first stimulated bone marrow-derived macrophages (BMDMs) with toll-like receptor 4 agonist (LPS for 24 h as a prototypical model of inflammatory (also called classical or M1) activation. Although Acly gene expression and protein levels remained unaffected, we observed a marked increase of phosphorylated Acly (p-Acly) in inflammatory macrophages (Fig. 1a, b). As these data suggested that Acly activation might support inflammatory responses we next examined Acly phosphorylation within human atherosclerotic plaques. Whereas total Acly was present in most cells in and around the plaque, activated (p-)Acly predominantly colocalized with CD45+CD68+ macrophages in human atherosclerotic plaques (Fig. 1c, d, Supplementary Fig. 1a–c). In addition, unstable plaques showed increased abundance of activated Acly when compared to stable plaques (Fig. 1e). Rupture-prone plaque areas are known to be dominated by inflammatory macrophages[19] and together with our observation that the levels of activated p-Acly were increased in inflammatory macrophages, this prompted us to study whether targeting macrophage Acly could improve atherosclerosis outcome.

**Macrophage Acly deletion stabilizes atherosclerotic plaques.** To examine the role of macrophage Acly in atherogenesis, we crossed Acly^fl/fl mice[20] with mice expressing Cre recombinase under control of the Lyz2 promoter/enhancer regions (LysMcre) to generate a new genetic mouse model that specifically lacks Acly in the myeloid cell lineage (macrophages, monocytes, and neutrophils). After confirming the knockdown of ACLY in macrophages, myeloid dendritic cells, and neutrophils from those mice (hereafter referred to as Acly^M-KO, Supplementary Fig. 1d, e), we transplanted lethally irradiated atherosclerosis-susceptible Ldlr-deficient mice with bone marrow from either Acly^M-KO mice or Cre-negative Acly^fl/fl littermate control mice. After validating efficient engraftment in both experimental groups (Supplementary Fig. 1f), the mice were fed a high-fat diet (HFD) (Fig. 2a). All mice exhibited a similar increase in body weight and plasma cholesterol, and similar end-state hepatic triglyceride levels, whereas the increase of plasma triglycerides was attenuated in Acly^M-KO mice (Supplementary Fig. 1g–j). Also, both groups showed a comparable blood leukocyte composition with only B-cell levels being slightly lower in Acly^M-KO mice (Supplementary Fig. 1k–o). As splenic B-cell levels remained unaltered, there is no apparent general defect in B-cell development or activation. It rather suggests an indirect, and yet unknown, effect of myeloid Acly knockdown on circulating B cells.

After 10 weeks of HFD, immunohistochemical and gene expression analysis on lesions confirmed Acly knockdown in myeloid cells within the atherosclerotic plaques of Acly^M-KO mice (Supplementary Fig. 2a, b). Although the atherosclerotic plaque size was similar, pathological scoring of plaques indicated an increase in thick fibrous cap atheromas in the Acly^M-KO group (Fig. 2b–d). The observation that the necrotic area was significantly smaller in Acly^M-KO-transplanted mice (Fig. 2e, f) indicated that the lesions of those mice are more stable as necrotic core formation is associated with plaque instability and rupture[5]. Moreover, analysis of Sirius red staining highlighted increased collagen content and fibrous cap thickness in plaques of

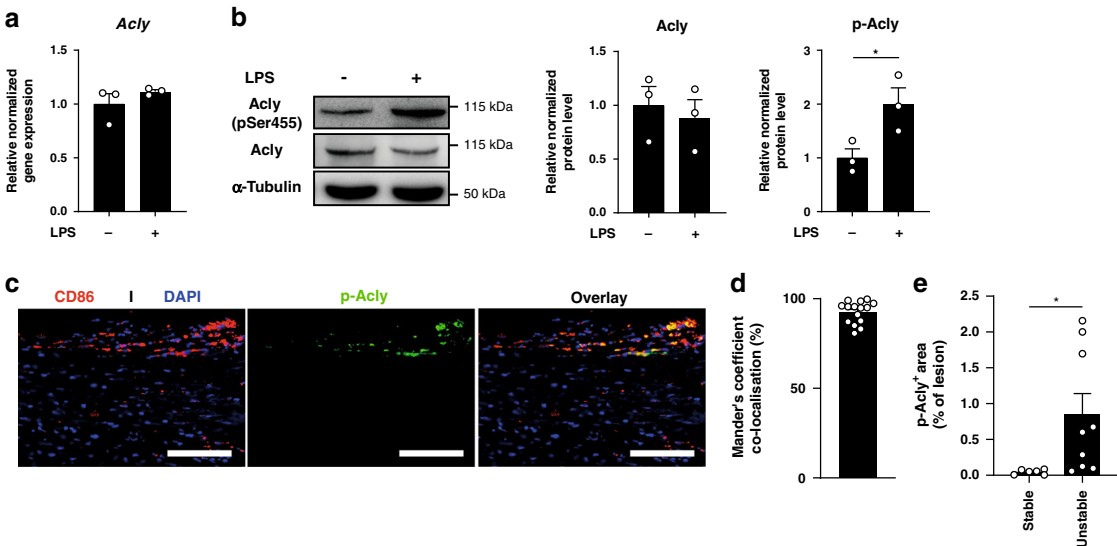

**Fig. 1 Acly is phosphorylated in inflammatory conditions in vitro and in vivo. a** Relative normalized expression of Acly in unstimulated and LPS-stimulated macrophages. **b** Normalized protein levels from cell lysates of unstimulated and LPS-stimulated macrophages. Samples were immunoblotted with antibodies against Acly, phosphorylated Acly (p-Acly) and α-tubulin. Acly/p-Acly quantification on the blots derive from samples of the same experiment and gels/blots were processed in parallel. *$P = 0.0441$. **c** Representative immunohistochemical staining for macrophages (CD68) and p-ACLY in human plaques from 16 stable/unstable plaques. Scale bar represents 100 μm. **d** Quantification of colocalization, percentage of p-Acly+ macrophages overlapping with CD68+ area. **e** Quantification of p-Acly+ area in the lesion. *$P = 0.0402$. Values represent mean ± SEM ($n = 3$ technical replicates of three pooled mice **a**, $n = 3$ one representative image of three technical replicates of three pooled mice (**b**, western blot), $n = 7/9$ stable/unstable plaques **d**, **e**. *$P < 0.05$; by two-tailed Student's $t$ test (**b**, **e**). Source data are provided as a Source Data file (**a-e**).

Acly^{M-KO}-transplanted mice (Fig. 2g–i), another sign of plaque stability. Immunohistochemical analysis revealed that the increase in collagen deposition was not accompanied by differences in macrophage and neutrophil content in the lesions (Fig. 2j, k, Supplementary Fig. 2c). Plasma triglyceride levels did not correlate with the plaque characteristics, indicating that lower triglyceride levels were not the cause of the more stable plaque phenotype (supplementary Fig. 2d). Transforming growth factor beta (TGF-β) is a key anti-atherogenic cytokine that can suppress immune effector functions and stabilizes atherosclerotic lesions by inducing fibrotic activities of smooth muscle cells and fibroblasts[21]. We therefore assessed its expression in aortic arches and detected increased *Tgfb1* expression in Acly^{M-KO}-transplanted mice (Fig. 2l). As labeling of TGF-β and the macrophage marker MOMA-2 suggested an increase of macrophage TGF-β but not total TGF-β (Fig. 2m–o), the increased *Tgfb1* levels were probably caused by macrophages within the lesion. Collectively, these in vivo data demonstrate that Acly^{M-KO}-transplanted mice had a favorable plaque phenotype, reflected by decreased necrosis and increased collagen deposition and fibrous cap thickness.

**Acly regulates in vitro macrophage polarization.** A possible mechanism to explain the more fibrotic plaque phenotype in the Acly^{M-KO} mice is a change in macrophage polarization. Although inflammatory (M1) macrophages can promote atherogenesis, IL-4-induced macrophages (M2) are regarded as atheroprotective because of their anti-inflammatory and pro-fibrotic homeostatic properties[1]. Previous in vitro studies with inhibitors suggested that Acly is involved in both inflammatory and IL-4-induced macrophage polarization[10,22,23]. Therefore, we next sought to reassess these findings with the aim to study whether altered macrophage polarization could clarify the differences in plaque phenotypes (Fig. 3a).

Distinct macrophage subsets differentially affect plaque phenotype. As previous inhibitor studies reported an involvement of Acly in both M1 and M2 macrophage polarization[10,11],

we next aimed to investigate whether altered macrophage polarization in the absence of Acly could clarify the differences in plaque phenotypes (Fig. 3a). Hereto, we first stimulated macrophages from control and Acly^{M-KO} mice with LPS to elicit an inflammatory macrophage response. Surprisingly, our genetic Acly^{M-KO} model indicated that Acly is not needed for inflammatory macrophage responses. The LPS-induced surface markers CD40, CD86, and major histocompatibility complex (MHC)-II were upregulated to the same extend in both groups and CD80 expression was higher in Acly^{M-KO} macrophages (Fig. 3b). Moreover, the LPS-induced levels of pro-inflammatory genes and factors such as IL-6, tumor necrosis factor (TNF), nitric oxide (NO), and reactive oxygen species (ROS) were even increased in the absence of Acly (Fig. 3c, d). Likewise, we detected increased inflammatory gene expression in vivo in the atherosclerotic lesions and peritoneal macrophages of the Acly^{M-KO} group (Fig. 3e, Supplementary Fig. 2e).

Next, we assessed the effect of Acly deficiency on IL-4-induced macrophage polarization (Fig. 4a). Hereto, we measured the IL-4-elicited expression of commonly used surface markers and detected reduced levels of CD206, CD273, and CD301 on the surface of Acly-deficient macrophages, along with blunted M2-associated gene expression (Fig. 4b, c). This decreased IL-4 response may be explained by lower histone 3 lysine 27 (H3K27) acetylation in the absence of Acly and confirms a previous study that applied Acly inhibitors during IL-4-responses in macrophages to link Acly-mediated production of acetyl-CoA to histone acetylation[10]. Conversely, H3K27 levels were similar in naive and LPS-treated control and Acly^{M-KO} macrophages and thus histone acetylation is not the mechanism explaining the macrophage phenotype in this setting (Supplementary Fig. 2f). Together, the enhanced inflammatory activation and reduced IL-4 responses in the absence of Acly do not explain the advantageous plaque phenotype of Acly^{M-KO}-transplanted mice. Our data highlight that M1/M2 polarization in vitro not necessarily reflects the in vivo context and underscore that caution is necessary when interpreting inhibitor effects.

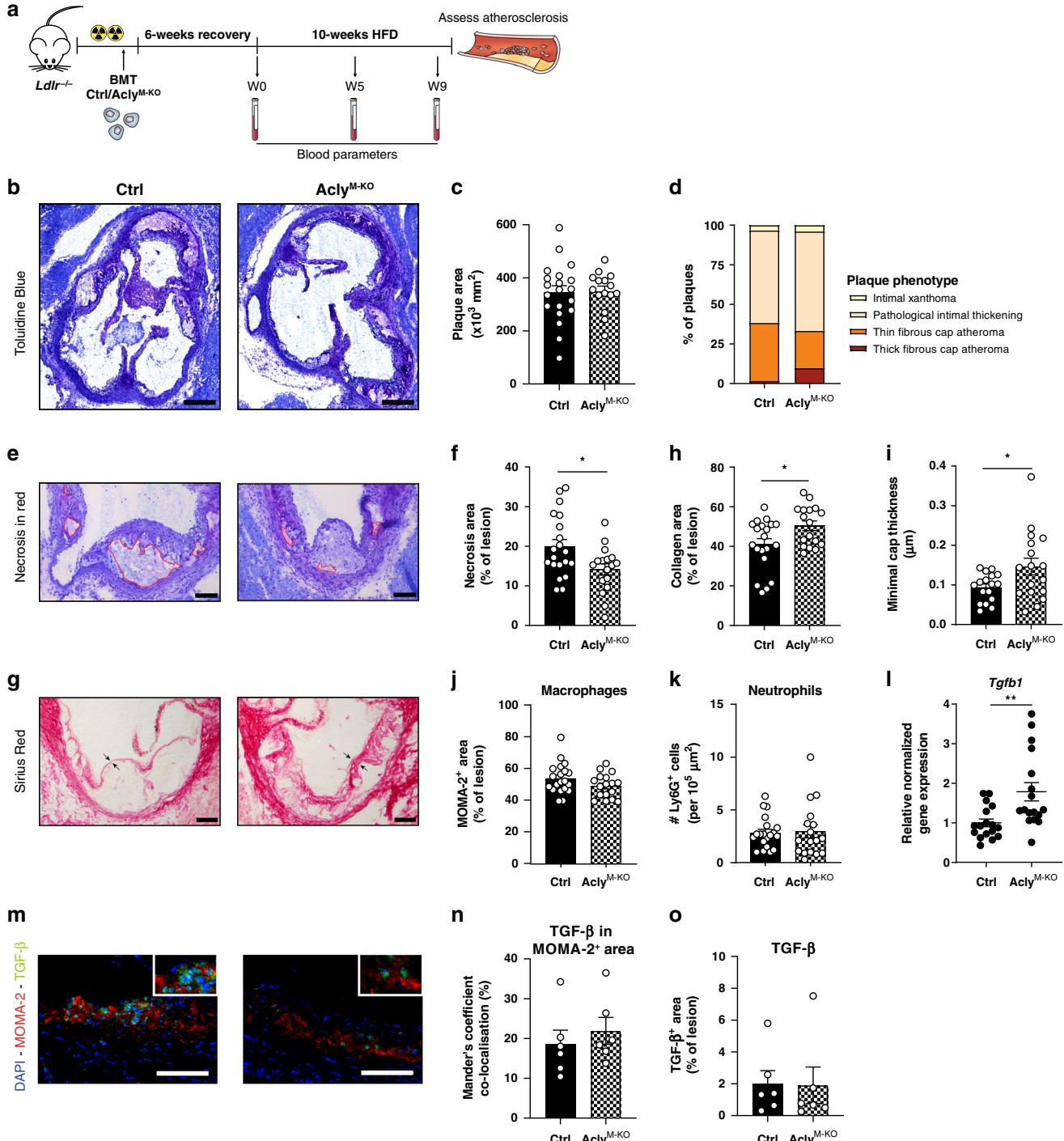

**Fig. 2 Acly deletion in macrophages elicits a favorable plaque phenotype. a** In order to asses atherogenesis, bone marrow cells of either control of Acly^M-KO were transplanted in lethally irritated Ldlr^−/− mice. Six weeks after transplantation, mice were put on a HFD for 10 weeks. **b** Representative toluidine blue staining of atherosclerotic plaques within the aortic roots of control and Acly^M-KO mice. Scale bar represents 500 μm. **c** Quantification of plaque size. **d** Plaque phenotypes as classified by an experimental pathologist. **e** Toluidine blue staining of atherosclerotic lesions with necrosis indicated in red. Scale bar represents 200 μm. **f** Quantification of the necrotic area. *P = 0.0173. **g** Sirius red staining of plaques. Scale bar represents 200 μm. **h** Quantification of the collagen deposition. *P = 0.0201. **i** Minimal cap thickness was measured at the thinnest region of the fibrotic cap surrounding the necrotic core as indicated by the arrows in **g**. *P = 0.0292. **j** Quantification of MOMA-2^+ area for macrophages. **k** Quantification of Ly6G^+area for neutrophils. **l** Normalized expression of Tgfb1 in aortic arches of control and Acly^M-KO mice. **P = 0.0038 **m** Representative immunohistochemical analysis of macrophages (MOMA-2) and TGF-β in mouse plaques from control or Acly^M-KO mice (n = 6/6 ctrl/KO from one experiment). Scale bar represents 100 μm. **n** Quantification of colocalization, percentage of TGF-β^+ area in MOMA-2^+ area. **o** Quantification of TGF-β^+ area in the lesion. Values represent mean ± SEM (n = 20/17 (ctrl/KO in **b–l**), n = 6/6 (ctrl/KO in **n**, **o**). *P < 0.05; **P < 0.01 by two-tailed Student's t test (**f**, **h**, **i**, **l**). Source data are provided as a Source Data file (**c**, **d**, **f**, **h–l**, **m**, **o**).

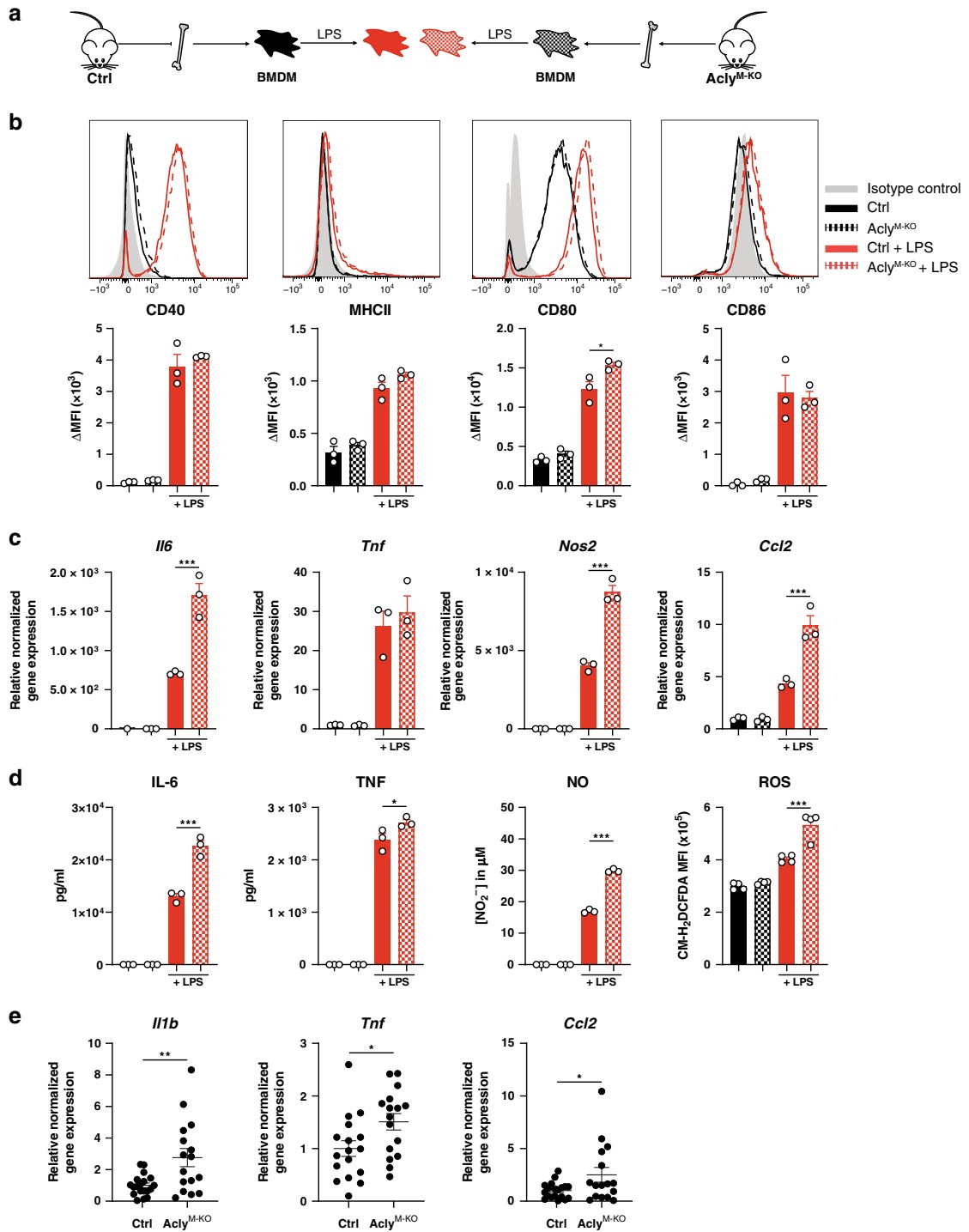

**Fig. 3 Altered macrophage polarization in vitro does not explain the plaque phenotype in vivo. a** Differentiated control and Acly$^{M-KO}$ macrophages (BMDMs) were stimulated with LPS. **b** Expression of LPS-induced surface markers as measured by flow cytometry. Representative histograms and quantified surface expression (ΔMFI = [median fluorescence intensity]$_{positive\ staining}$ − [MFI]$_{pooled\ control}$) are shown. CD80 *$P$ = 0.0160. **c** Relative normalized gene expression of indicated LPS-induced genes. *Il6* ***$P$ < 0.0001, *Nos2* ***$P$ < 0.0001, *Ccl2* ***$P$ = 0.0002. **d** Production of IL-6 ***$P$ < 0.0001, TNF *$P$ = 0.0281, NO ***$P$ < 0.0001 and ROS ***$P$ < 0.0001. **e** Normalized expression of M1 genes in aortic arches of control and Acly$^{M-KO}$-transplanted mice. *Il1b* **$P$ = 0.0030, *Tnf* *$P$ = 0.0183, *Ccl2* *$P$ = 0.0321. Values represent mean±SEM of $n$ = 3 (**b–d**) technical replicates of one out of three representative experiments or n = 20/17 (ctrl/KO in **e**). *$P$ < 0.05; **$P$ < 0.01, ***$P$ < 0.001 by ordinary one-way ANOVA with Bonferroni post hoc test for multiple comparisons (**b–d**) or two-tailed Student's $t$ test (**e**). Source data are provided as a Source Data file (**b–e**).

**Acly does not regulate core metabolic processes.** To identify the mechanism by which myeloid deletion of *Acly* influences macrophage and plaque phenotype, we explored changes in core metabolic pathways as a potential explanation of the increased inflammatory response in Acly-deficient macrophages.

Metabolomics revealed that the expected LPS-induced accumulation of citrate was more pronounced in the absence of Acly, signifying decreased Acly-mediated conversion of citrate into acetyl-CoA[24,25]. Yet, loss of Acly led to no further changes in the abundance of other tricarboxylic acid cycle (TCA) cycle

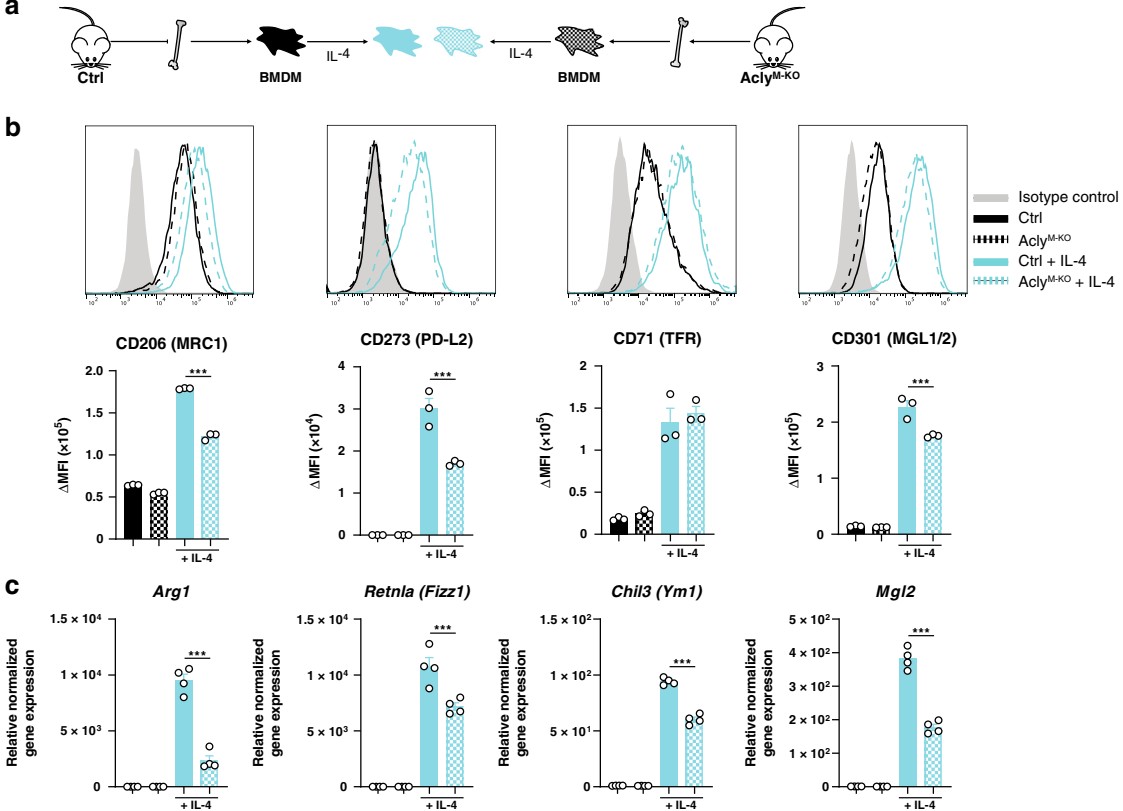

**Fig. 4 IL-4 response is decreased after Acly deletion in macrophages. a** Differentiated control and Acly[M-KO] macrophages (BMDMs) were stimulated with IL-4. **b** Expression of IL-4-induced surface markers as measured by flow cytometry. Representative histograms and quantified surface expression ($\Delta$MFI = [median fluorescence intensity]$_{positive\ staining}$ – [MFI]$_{pooled\ control}$) are shown. CD206 ***$P < 0.0001$, CD273 ***$P = 0.0003$, CD301 ***$P = 0.0008$. **c** Relative normalized gene expression of IL-4-induced genes. *Arg1* ***$P < 0.0001$, *Retnla* ***$P = 0.0004$, *Ym1* ***$P < 0.0001$. Values represent mean ± SEM of $n = 3$ **b** or $n = 4$ **c** technical replicates of one out of three representative experiments. ***$P < 0.001$ by ordinary one-way ANOVA with Bonferroni post hoc test for multiple comparisons (**b**, **c**). Source data are provided as a Source Data file (**b**, **c**).

intermediates (Fig. 5b). Extracellular flux analysis, and glucose, lactate and ATP measurements showed no differences in mitochondrial function, glycolysis and ATP production, indicating that other mechanisms should explain the observed inflammatory phenotypes of Acly-deficient macrophages (Fig. 5c–g).

**Deregulated cholesterol metabolism in the absence of Acly.** To reveal such mechanism, we next performed RNA-sequencing on BMDMs from control and Acly[M-KO] mice. Plotting genes in a volcano plot and performing pathway analysis on the differentially expressed genes identified cholesterol biosynthesis as the most deregulated pathway, with most targets related to this pathway being upregulated in Acly[M-KO] macrophages (Fig. 6a, b). In addition to genes involved in cholesterol metabolism,[10] fatty-acid metabolism, and cell cycle genes were differentially expressed. Distinct genes involved in cholesterol and fatty-acid import and transport (*Ldlr, Cd36, Scarb1, Fabp3, Slc27a4*) were increased in the absence of Acly. In contrast, Acly[M-KO] macrophages showed decreased expression of LXR-regulated genes that promote cholesterol efflux (*Abca1, Abcg1, Apoe*) or limit cholesterol uptake (*Mylip*) (Fig. 6c). The most significantly increased gene in Acly[M-KO] macrophages was *Dhcr24*, encoding the enzyme that converts desmosterol into cholesterol as the terminal step in cholesterol synthesis (Fig. 6d). In line with increased *Dhcr24* expression, we measured lower levels of desmosterol and desmosterol-induced genes in Acly[M-KO] macrophages (Fig. 6e, f). By increasing cholesterol synthesis and import, and via limiting

its export, Acly-deficient macrophages possibly try to cope with the reduced supply of Acly-mediated acetyl-CoA, and by doing so they manage to secure total cholesterol levels (Fig. 6g). Moreover, Acly-deficient macrophages might rescue acetyl-CoA production by upregulating *Acss2* (Fig. 6h). This gene encodes acyl-coenzyme A synthetase short-chain family member 2, an enzyme that converts acetate into acetyl-CoA. This is in line with earlier reports that describe induction of *Acss2* in adipocytes and mouse embryonic fibroblasts upon Acly deletion[20,26].

The resulting deregulated cholesterol metabolism and suppressed desmosterol levels could explain the lower expression of LXR-target genes (Fig. 6f) and possibly also altered polarization (Figs. 3, 4) in Acly[M-KO] macrophages. Indeed, LXR activation is involved in reprogramming fatty-acid metabolism and inhibiting LPS-induced inflammatory-response genes in macrophages[27]. Supporting this hypothesis, exposing macrophages to LXR agonist GW3965 decreased inflammatory responses in both control and Acly[M-KO] macrophages (Supplementary Fig. 3a). As such, reduced LXR activation could provide one potential mechanistic explanation for the elevated inflammatory responses of *Acly*-deficient macrophages.

**Fatty-acid synthesis is affected in the absence of Acly.** Apart from cholesterol metabolism, reactome pathway analysis also indicated "fatty acyl-CoA biosynthesis" and "fatty-acid metabolism" as top deregulated pathways (Fig. 6b). As Acly provides acetyl-CoA as a precursor for fatty acid synthesis[28,29], we

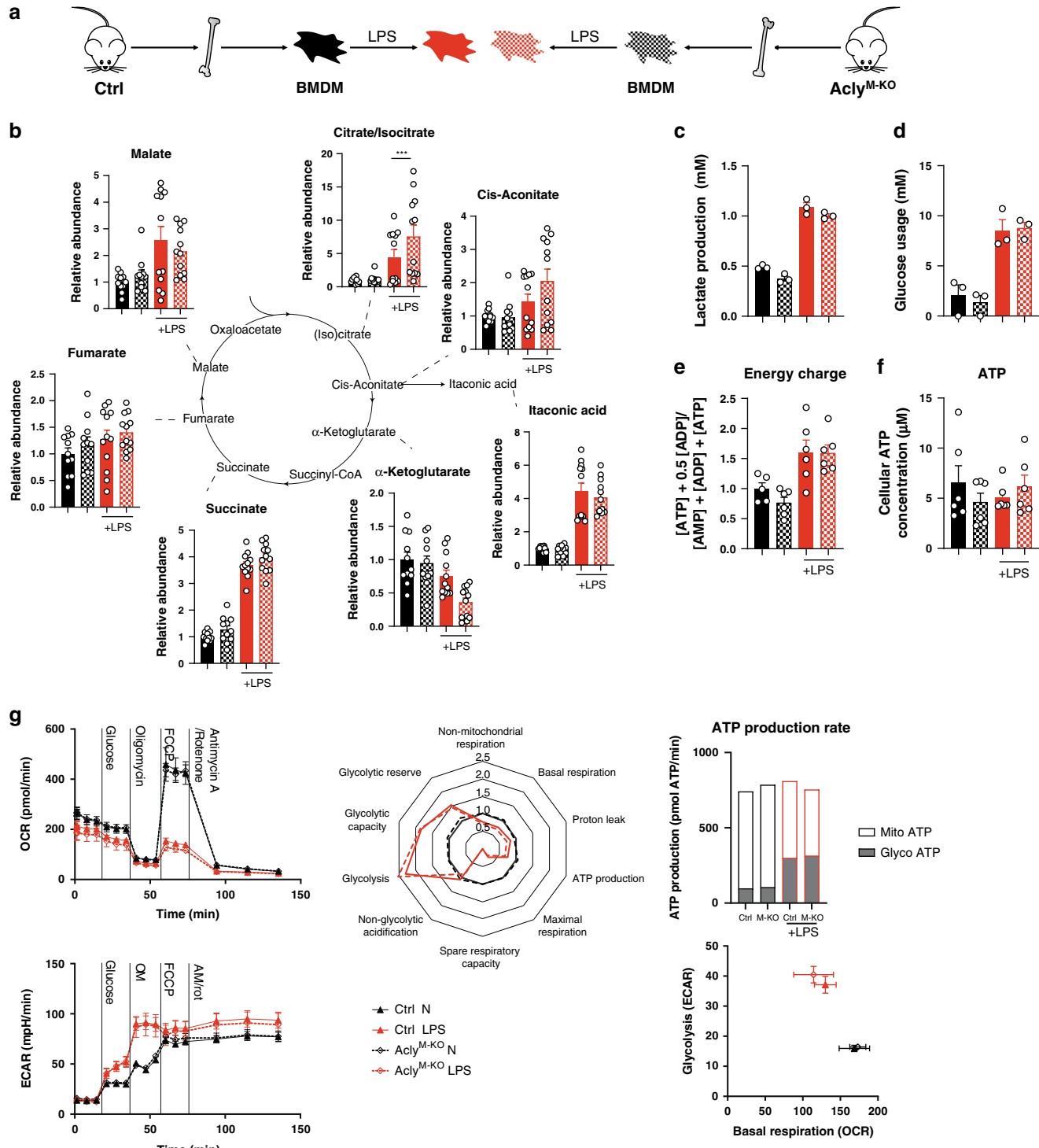

**Fig. 5 Mitochondrial function remains unaltered in Acly$^{M-KO}$ BMDMs. a** Differentiated control and AclyM-KO macrophages (BMDMs) were left untreated or stimulated with LPS (**b**) relative abundance of TCA cycle intermediates ***$P < 0.0001$. **c** Lactate secretion after 24 h LPS treatment. **d** Metabolized glucose in 24 h. **e** Cellular energy charge as calculated by [ATP] + .05[ADP]/([AMP] + [ADP] + [ATP]). **f** Cellular ATP concentration. **g** Seahorse analyses and extracted parameters. Values represent mean ± SEM from $n = 12$ technical replicates from two combined experiments **b**, $n = 3$ technical replicates from three pooled mice from one of three representative experiments (**c, d, g**), $n = 6$ technical replicates from three pooled mice from one of two representative experiments (**e, f**). ***$P < 0.001$ by two-way ANOVA with Bonferroni post hoc test for multiple comparisons. Source data are deposited in MTBLS2159 (**b**) or provided as a Source Data file (**c–g**).

measured fatty acid levels in control and Acly$^{M-KO}$ macrophages by performing lipidomics measurements. Whereas RNA-sequencing analysis suggested a lipid-laden phenotype of Acly$^{M-KO}$ macrophages, total lipid pools were similar in control and Acly$^{M-KO}$ macrophages (Fig. 7a, b). Conversely, levels of omega 3 and 6 fatty acids, including arachidonic acid (AA) tended to be increased in Acly$^{M-KO}$ macrophages (Fig. 7c, d). Cyclooxygenase (COX)-1 and COX-2 produce prostaglandins

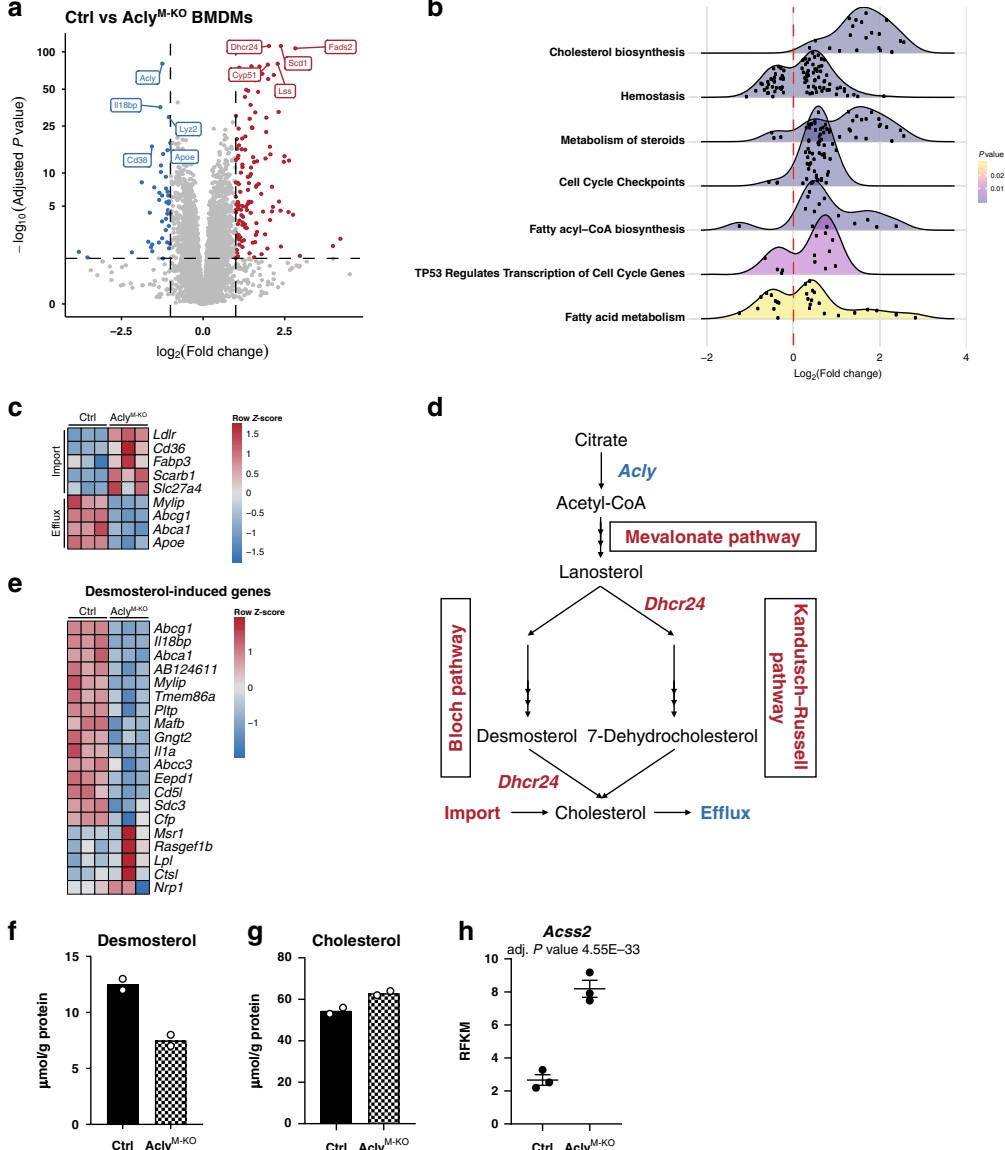

**Fig. 6 Acly[M-KO] macrophages exhibit deregulated cholesterol biosynthesis. a** Volcano plot with up- or downregulated genes in *Acly*-deficient macrophages (BMDMs) highlighted in red or blue, respectively. **b** Ridgeline plot of significantly enriched Reactome pathways. Log2FC distributions of member genes are plotted on the *x* axis. Color of the distributions represents enriched pathway *P* value. **c** Expression of genes involved in import/transport of cholesterol and fatty acids, and genes that promote cholesterol efflux. **d** Schematic view of cholesterol metabolism and the effect of *Acly* deficiency (red indicating upregulation; blue downregulation). **e** Desmosterol levels in control and Acly-deficient macrophages. **f** Genes previously found to be induced by desmosterol, plotted in a heatmap for control and Acly[M-KO] macrophages. **g** Cholesterol levels in control and Acly-deficient macrophages. **h** *Acss2* expression in control and Acly-deficient macrophages. Values represent mean from *n* = 2 technical replicates from three pooled mice **e**, **g**, or mean ± SEM *n* = 3 technical replicates from three pooled mice (**h**). Source data are provided as a Source Data file **e**, **g** or deposited in GSE126690 (**a–d**, **f**, **h**).

from AA and these lipid mediators are known to affect inflammatory responses[30]. The genes encoding COX-1 and COX-2 (*Ptgs1* and *Ptgs2*) and downstream prostaglandins (PGD₂, PGF₂ₐ, and in particular PGE₂) were deregulated in Acly[M-KO] macrophages (Fig. 7d). Moreover, Acly[M-KO] macrophages showed a decreased expression of prostaglandin E 4 receptor (EP4, *Ptger4*) (Fig. 7e). Loss of EP4 has previously been described to enhance inflammatory responses[31,32]. Subsequently, PGE₂- and LPS-treated Acly[M-KO] macrophages display a reduced responsiveness to PGE₂ when compared to control macrophages (Fig. 7f, g).

Together, these findings indicate that Acly[M-KO] macrophages rewire cholesterol and fatty-acid metabolism to deal with the absence of Acly, and these metabolic changes are accompanied by altered inflammatory responses.

**Acly[M-KO] macrophages show increased apoptosis and efferocytosis.** In addition to cholesterol and fatty acid metabolism, pathway analysis identified "cell cycle checkpoints" and "TP53 regulates transcription of cell cycle genes" among the most-regulated pathways (Fig. 6b), with most genes related to these pathways being increased in Acly[M-KO] macrophages (Fig. 8a). As the transcription factor p53 is a key modulator of both apoptosis and senescence, we assessed those cell cycle-related processes more detailed. We stained DNA of control and Acly[M-KO] macrophages with propidium iodide and measured increased fractional DNA content (Sub-G1), a hallmark of ACs, in Acly[M-KO] macrophages (Fig. 8b). Moreover, β-galactosidase activity measurements indicated increased senescence in Acly[M-KO] macrophages (Fig. 8c). Supporting those data, the expression of p53

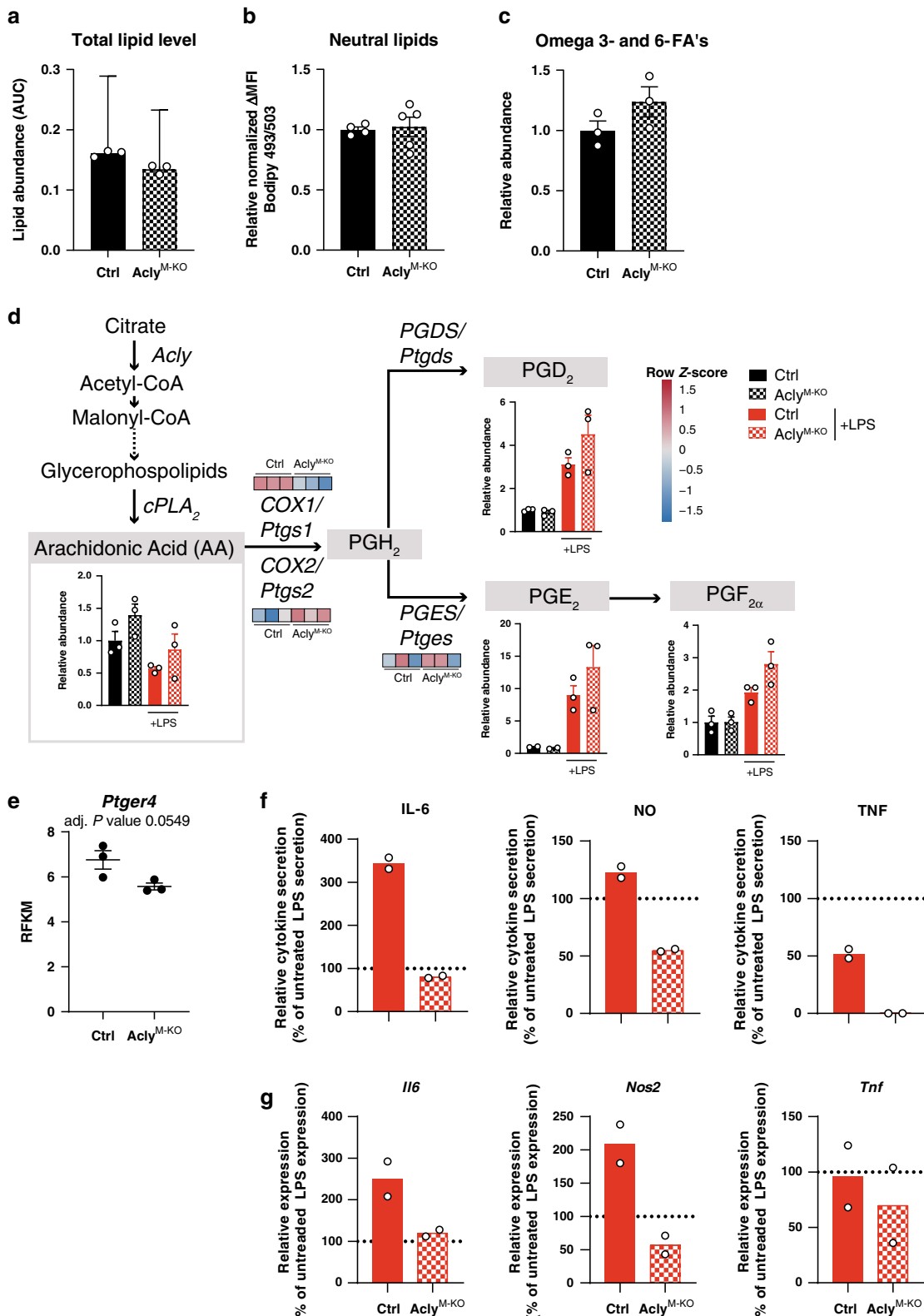

**Fig. 7 Deletion of Acly leads to deregulation of lipid mediators and decreased responsiveness to prostaglandin E2. a** Total sum of measured lipids by metabololipidomics. **b** Neutral lipids analyzed by Bodipy 493/503. **c** Sum of AA, DHA, EPA, AdA, LA, ALA/GLA, DPAn-3, and DGLA omega 3 and 6 fatty acids (FA's) measured by metabololipidomics. **d** Lipid abundances and genes involved in prostaglandin synthesis. **e** *Ptger4* expression in control and Acly-deficient macrophages. **f** Effect of PGE2 treatment on cytokine secretion in control and Acly^M-KO LPS-treated BMDMs. **g** Effect of PGE2 treatment on cytokine secretion in control and Acly^M-KO LPS-treated BMDMs. Values represent mean ± SEM ($n = 3$ technical replicates from three pooled mice **a–e**), or mean ($n = 2$ technical replicates from three pooled mice (**f**, **g**)) Source data are provided as a Source Data file (**a–d**, **f**) or deposited in GSE126690 (**d**, **e**).

target genes related to cell cycle mostly increased in the absence of Acly (Fig. 8a). Noteworthy, the expression of the LXR-dependent anti-apoptosis genes *Cd5l* (AIM), *Lbp*, and *Mafb* (all known to promote macrophage survival[33–35]) was reduced in the absence of Acly, providing a potential link between reduced LXR activation (Fig. 6f) and increased apoptosis in these cells. Increased apoptosis within atherosclerotic lesions needs to be paralleled by increased clearance of those cells to prevent the growth of necrotic cores and plaque destabilization. Therefore, we questioned whether improved efferocytosis of apoptotic macrophages could explain the smaller necrotic cores in atherosclerotic lesions of Acly[M-KO] mice. Hereto, we co-cultured control or Acly[M-KO] BMDMs with apoptotic macrophages that were labeled with the pH-sensitive dye pHrodo and measured efferocytosis. Culturing the cells at 4 °C served as a negative control to set the pHrodo[+] gate and effective internalization of ACs was also validated by imaging flow cytometry (Fig. 8d). To validate these results in vivo, we analyzed apoptosis and efferocytosis in the atherosclerotic plaques from control and Acly[M-KO] mice by terminal deoxynucleotidyl transferase dUTP nick end labeling (TUNEL) double staining with MAC3. In vivo apoptosis, but not efferocytosis, appeared to be increased in plaques from Acly[M-KO] mice (Fig. 8e, f). These assays highlighted that Acly-deficient macrophages demonstrate more-efficient efferocytosis in vitro, an atheroprotective macrophage function associated with stabilized atherosclerotic plaques in vivo (Fig. 8d).

## Discussion

Targeting ACLY in liver cells has emerged as an attractive aid to lower serum LDL-C levels in hypercholesteremic patients[16,17]. In the present study, we demonstrate that myeloid deficiency for Acly deregulated fatty-acid and cholesterol metabolism and leads to enhanced apoptosis, combined with improved efferocytosis. In vivo, this results in more stable atherosclerotic plaques, reflected by smaller necrotic cores and more collagen deposition. As such we identify targeting macrophage Acly as a potential new therapeutic avenue in cardiovascular disease.

Noteworthy, Acly targeting with bempedoic acid (ETC-1002, a prodrug activated by acyl-CoA synthetase solute carrier family 27 member 2 (SLC27A2) in the liver) is already applied in the clinic as a new aid to lower serum LDL-C in hypercholesteremic patients[36,37]. Acly inhibition reduces cholesterol synthesis in hepatocytes, leading to the upregulation of LDLR expression and clearance of LDL-C particles from the blood[38]. Similar as in liver cells, Acly deficiency in macrophages leads to the upregulation of *Ldlr*, probably to compensate for deregulated cholesterol biosynthesis. In addition to elevated expression of the LDL receptor, loss of *Acly* enhances the expression of CD36 and SR-B1. Besides their role in cholesterol uptake, both class B scavenger receptors are key mediators of efferocytosis[39,40]. Impaired efferocytosis drives the accumulation of necrotic debris in atherosclerotic plaques and can lead to plaque rupture and cardiovascular events. In accordance with increased CD36 and SR-B1 expression in Acly[M-KO] macrophages, the capacity of these cells to clear ACs was improved. Silencing of ACLY was recently found to suppress DNA damage repair, resulting in genomic instability and cell death[41]. In agreement, *Acly*-deficient macrophages were more senescent and prone to apoptosis. Together, increased apoptosis and more effective clearance of such apoptotic macrophages may explain the reduced necrotic core formation in plaques of Acly[M-KO] mice. Along with reduced necrotic core formation, collagen deposition was increased in the plaques of Acly[M-KO]-transplanted mice. In line with the increase in collagen content and fibrous cap thickness, the plaques of those mice showed increased mRNA levels of TGF-β1, a key pro-fibrotic cytokine which is secreted by

macrophages upon efferocytosis[42]. Thus, efferocytosis might trigger the synthesis of TGF-β1 that drives the increased collagen deposition in plaques of myeloid Acly-deficient mice. Interestingly, although efferocytosis is generally believed to be an anti-inflammatory process, we observed augmented expression of inflammatory genes in plaques of Acly[M-KO] mice. Moreover, cultured Acly-deficient macrophages were found to be hyperinflammatory. We found that the expression of LXR-target genes and abundance of main LXR agonist desmosterol were diminished in both unstimulated and LPS-stimulated Acly-deficient macrophages. In addition to their role in cholesterol efflux, LXRs are established negative regulators of inflammatory macrophage responses[43]. Therefore, blunted LXR activation provides a potential mechanistic explanation for the elevated inflammatory responses of Acly-deficient macrophages. In addition, LXR activation is believed to promote macrophage survival[33,35]. LXR-target genes *Cd5l* (AIM), *Lbp* and *Mafb* have been found to inhibit apoptosis and promote macrophage survival and were all downregulated in Acly-deficient macrophages. Therefore, diminished LXR activation might contribute to the increase in apoptosis upon Acly deletion. Interestingly, Lauterbach et al.[11] recently applied Acly inhibitors and siRNA-mediated knockdown to demonstrate the role of Acly in regulating acute LPS responses via histone acetylation. The difference between our observations and the ones of Lauterbach et al. might be explained by the knockdown method (i.e., short-term siRNA vs long-term LysMCre-driven) and other experimental variables. Clearly, the role of Acly in macrophages is stimulus- and time-dependent and also appears to depend on the experimental setting as Namgaladze et al. observed no effect of Acly silencing[44]. As such, this is an area that warrants future research.

Collectively, we demonstrate that macrophage-specific targeting of *Acly* elicits a favorable atherosclerotic plaque phenotype. With liver-specific inhibition of ACLY already being applied as a cholesterol-lowering aid, we anticipate our observations in macrophages to spike the development of new cell-specific ACLY inhibitors that target different cell types to tackle the disease from distinct angles.

## Methods

**Human tissue collection.** Anonymised human plaque tissues for the determination of protein expression were obtained from the Maastricht Pathology Tissue Collection (MPTC) under approval of the Medical Ethical Committees AZm/UM from the Maastricht University Medical Centre. The formalin-fixed paraffin-embedded human carotid samples used for immunohistochemistry were collected at autopsy (n = 16, mean age 72.6 years, all men). Collection, storage in the MPTC and use of tissues were performed in agreement with the "Dutch Code for Proper Secondary use of Human Tissue". None of the co-authors was involved in tissue collection during autopsy. No prior informed consent was directly obtained from the patients, but an opt-out arrangement was in place and hence tissues were not used in case of objection by relatives in line with the guidelines provided by the Dutch Code for Proper Secondary use of Human Tissue. Samples represented the following stages of atherosclerosis (n = 7/9 per. respectively, stable/unstable group): pathological intimal thickening and plaque with intraplaque hemorrhage[45].

**Mice experiments.** C57Bl/6J mice with *loxP* sites flanking exon 9 of the *Acly* gene (*Acly*[fl/fl])[20] were crossed with *Lyz2*-Cre transgenic mice to generate mice with a myeloid-specific deletion of *Acly* (Acly[M-KO]). *Ldlr*[−/−] mice on a C57BL/6 background were purchased from Jackson Laboratories. All mice were maintained in an animal facility at an ambient temperature of 20–24 °C with 40–70% relative humidity under a 12:12 h light–dark cycle. All mouse experiments were conducted after approval by the Committee for Animal Welfare (University of Amsterdam and VU university Amsterdam).

For bone marrow transplantation, female *Lldr*[−/−] mice were lethally irradiated with six Gray on 2 successive days. After randomization based on weight, mice were divided in two groups of 20 and injected intravenously with 5 × 10[6] bone marrow cells from either control or Acly[M-KO] mice. From one week before until 5 weeks after transplantation, recipient *Lldr*[−/−] mice received autoclaved tap water containing antibiotics (100 mg/L neomycin and 60,000 U/l polymyxin B sulfate). Bone marrow transplantation efficiency was evaluated by qPCR for *Ldlr* on DNA extracted from blood. Six weeks after the transplantation, mice received a HFD)

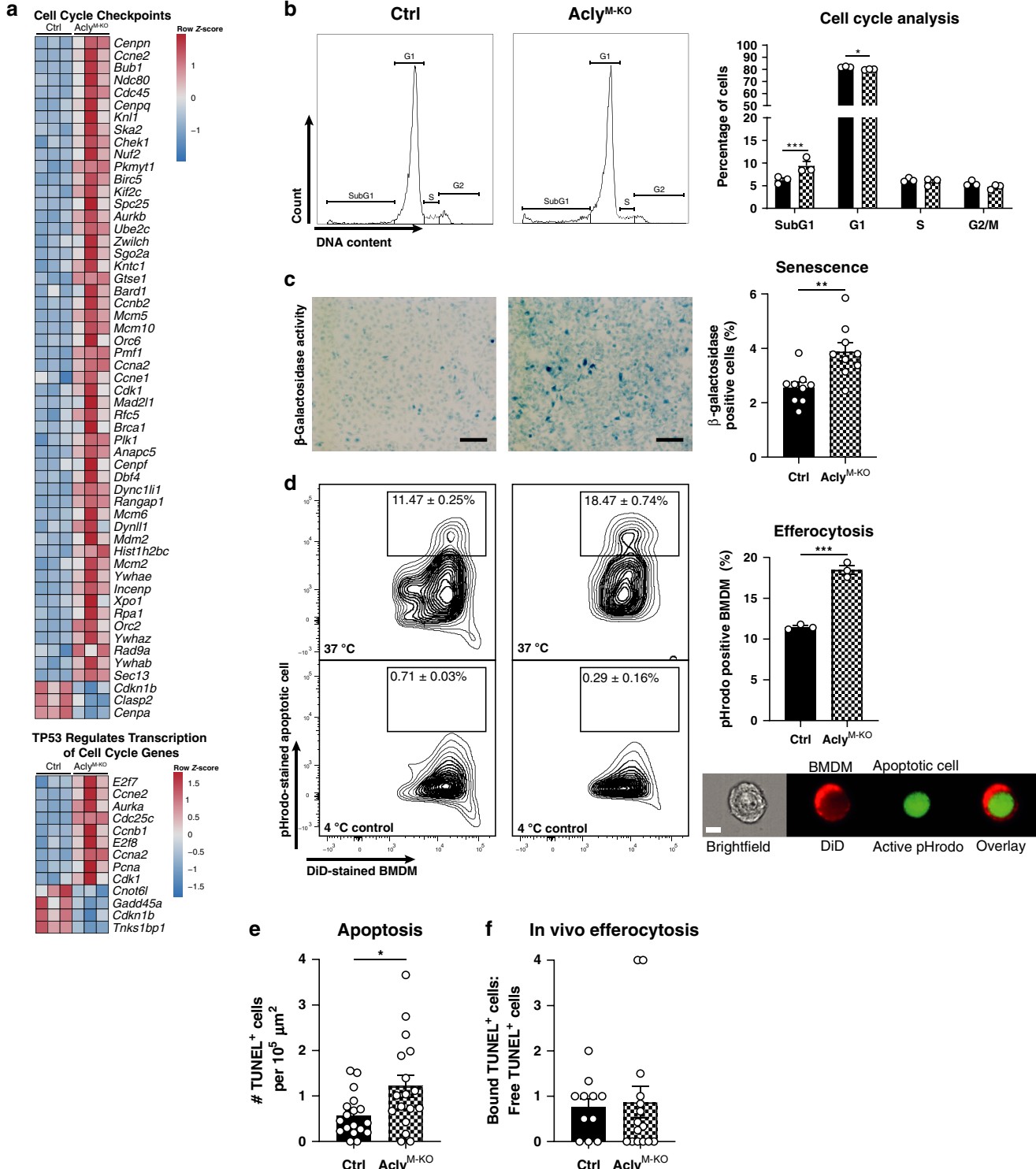

**Fig. 8 Deletion of Acly results in increased apoptosis, growth arrest, and more-efficient efferocytosis. a** Expression of genes associated with indicated reactome pathways. **b** Flow cytometric analysis of control and Acly$^{M-KO}$ macrophages stained with propidium iodide to asses cell cycle distribution. ***$P =$ 0.0009, *$P = 0.0385$. **c** Senescence in control and Acly$^{M-KO}$ BMDMs by b-galactosidase activity measurement. Scale bar represents 100 μm. **$P =$ 0.0034. **d** In vitro phagocytosis of pHrodo-stained apoptotic cells by DiD-labeled control or Acly$^{M-KO}$ BMDMs measured by flow cytometry after incubation at 37 °C (or 4 °C as negative control) and validated by imaging flow cytometry (Imagestream, scale bar represents 7 μm). Quantification of efferocytosis at 37 °C as a percentage of DiD-positive cells that incorporated an apoptotic cell. ***$P = 0.0002$. **e** Quantification of TUNEL$^+$ cells in lesions of Ldlr$^{-/-}$ mice transplanted with control or Acly$^{M-KO}$ bone marrow. *$P = 0.0203$ (**f**). Ratio of TUNEL$^+$ cells bound to MAC3$^+$ macrophages and free TUNEL$^+$ cells. Values represent mean ± SEM ($n = 3$ technical replicates of one out of three representative independent experiments in **b**–**d**) and $n = 20/19$ (ctrl/KO in **e**, **f**)). *$P < 0.05$; **$P < 0.01$, ***$P < 0.001$ by two-way ANOVA with Bonferroni post hoc test for multiple comparisons **b** or two-tailed Student's $t$ test (**c**, **d**). Source data are provided as a Source Data file (**b**–**f**) or deposited in GSE126690 (**a**).

(15% fat and 0.15% cholesterol, Ssniff Spezialdiäten). After 10 weeks HFD, mice were sacrificed by $CO_2$ asphyxiation. Hearts, livers, and spleens from killed mice were collected and hearts and livers were frozen in tissue-tek (DAKO) for histological analysis. Blood was collected before the start of the diet and five and nine weeks after the start of the diet for lipid profiling and to determine leukocyte composition. Peritoneal macrophages were isolated after thioglycolate injection from 6/6 (control/Acly$^{M-KO}$) mice as previously described[46].

**Histology.** Sections of 7 μm were cut on a Leica 3050 cryostat and stained with toluidine blue (0.2% in phosphate-buffered saline (PBS)) to quantify plaque area and Sirius red (0.05% in saturated picric acid solution) to measure collagen deposition using Adobe Photoshop software (Version. In addition, toluidine blue-stained slides were used to measure necrosis and to categorize plaques by an experienced pathologist. Minimal cap thickness was measured on the thinness part of the cap surrounding the necrotic core. Sections were fixed in acetone and incubated with antibodies against MOMA-2 (1:4000, AbD Serotec), Ly6G (1:200, BD Pharmingen), total Acly (1:100, Abcam), or MOMA-2 (1:4000, AbD Serotec) together with TGF-β (1:100, Abcam) to determine macrophage, neutrophil, total Acly and macrophage-related TGF-β content, respectively, (antibodies detailed in the Supplementary Table 1). MOMA-2, Ly6G, and Acly were captured using biotin-labeled rabbit anti-rat antibody (Vector Laboratories) as secondary antibody. Hereafter, the signal was amplified using the ABC kit (Vector Laboratories) and subsequently visualized using the ImmPACT AMEC Red substrate kit (Vector Laboratories). Combined MOMA-2 and TGF-β staining was visualized using Alexa-555-labeled goat anti-rat antibody (Molecular probes, 1:400) for MOMA-2 and Alexa647-labeled goat anti-rabbit antibody (Molecular probes, 1:400) for TGF-β. Hereafter, the signal was measured by multispectral imaging using a Vectra Polaris slide imager (Perkin Elmer), and subsequently cleaned from autofluorescent spectra by measuring an unstained lesion. ImageJ was used for analysis of positive area and for colocalization with the ImageJ plugin JaCOP.

To analyze apoptosis and efferocytosis with TUNEL, sections were stained with MAC3 following paraformaldehyde fixation, and permeabilization with 0.1% triton X-100. MAC3 signal was detected using biotinylated rabbit anti-rat antibody, streptavidin-ABC-AP, and visualized using vector blue (all Vector Laboratories). Next, TUNEL enzyme incorporated fluorescein-labeled dNTPs into ACs which were then visualized with anti-FITC–HRP, and diamino benzidine (DAB, DAKO) according to manufacturer's instruction (In situ cell death detection kit, Sigma). The efferocytosis index was calculated as the ratio of TUNEL + AC bound to MAC3 + cells over free ACs (bound AC/free AC), as previously described[47,48].

For the analysis of Acly in human plaques, slides were deparaffinised and antigens were retrieved by cooking in a Tris-EDTA buffer with pH 9. Plaque slides were incubated with anti-phosphorylated ATP citrate lyase (pSer$^{454}$) (Sigma, 1:100) or anti-ATP citrate lyase (Abcam, 1:100), anti-CD68 (Abcam, 1:75) or anti-CD45 (DAKO, 1:100), and DAPI (Invitrogen, 1:1000) to visualize macrophages and phosphorylation of ATP citrate lyase in the plaque tissue. Subsequently, slides were incubated with anti-rabbit Alexa488 (Molecular probes, 1:400) and anti-mouse Alexa647 (Invitrogen, 1:400). Fluorescent signal was captured with multispectral imaging using a Vectra Polaris slide imager (Perkin Elmer). ImageJ was used for analysis of positive area and for colocalization with the ImageJ plugin JaCOP. A representative image of 7 unstable plaques was shown.

**Blood, spleen and liver parameters.** Levels of cholesterol and triglycerides in plasma were enzymatically determined according to the manufacturer's protocol (Roche). Hepatic triglyceride levels were enzymatically determined on snap frozen liver samples according to the manufacturer's protocol (Cayman chemical). Spleens were crushed and digested by enzymatic digestion. Blood and spleen leukocyte composition was assessed by flow cytometry after antibody staining with CD45-APC-Cy7 (1:100) for total leukocytes, CD11b-PE-Cy7 (1:400), MHC-II-PerCP Cy5.5 (1:100) and Ly6C-APC (1:100) for monocytes, Ly6G-FITC (1:100) for neutrophils, NK1.1-PE (1:100) for natural killer cells, CD3-PerCP Cy5.5 (1:100), CD4-APC (1:200) and CD8-FITC (1:200) for T cells and CD19-PE (1:100) for B cells. Fluorescence was measured on a Beckman Coulter CytoFLEX and analyzed with CytExpert software.

**Bone marrow isolation/neutrophil isolation and BMDM/BMDC culture.** Bone marrow cells were extracted by flushing femurs and tibias of control and Acly$^{M-KO}$ mice. Cells were seeded and differentiated to macrophages in complete Roswell Park Memorial Institute (RPMI)-1640 containing 25 mM HEPES, 2 mM L-glutamine, 10% FCS, 100 U/ml penicillin, 100 μg/ml streptomycin and 15% L929-conditioned (LCM) medium for 7 days. BMDMs were left unstimulated or stimulated with 100 ng/ml LPS (Sigma) or 20 ng/ml IL-4 (Peprotech) for 24 h.

Bone marrow cells were differentiated to dendritic cells in IMDM containing 10% FCS, 100 U/ml penicillin, 100 μg/ml streptomycin, 2 mM Glutamax, 50 μM β-mercaptoethanol and 30 ng/ml X63 for 7 days.

Neutrophils were isolated from bone marrow cells by density gradient as also detailed in the reference[49]. Bone marrow cells were resuspended as max $100 \times 10^6$ per ml ice-cold PBS, after which 1 ml cell suspension was added on top of a double density gradient. The lower layer contained 3 ml of 150 μM NaCl-Percoll mix with

a density of 1.119 g/ml topped up with a layer of Lymphoprep with a density of 1.077 g/ml. After centrifugation (30 min, 2000 rpm, 25 °C, no brake) neutrophils were present at the interphase. After washing with RPMI, viability was determined and cells were frozen until further analysis.

**Cell lines.** RAW264.7 cells were cultured in complete RPMI-1640 containing 2 mM L-glutamine, 10% FCS, 100 U/ml penicillin and 100 μg/ml streptomycin. Cells were passed when 80–90% confluency was reached by scraping in fresh medium. Cells with passage number 6–12 were used for experiments.

**Immunoblotting.** BMDMs were washed with cold PBS and lysed on ice in NP40 cell lysis buffer (ThermoFisher) supplemented with fresh protease inhibitor cocktail (Sigma-Aldrich) and fresh PhosSTOP (Sigma-Aldrich). Lysates were scraped, collected and centrifuged for 10 min at 2000 × g and 4 °C. Supernatants were equalized on protein concentration after quantification with the BCA assay (ThermoFisher), heated for 10 min at 95 °C, loaded on Bolt 4–12% Bis-Tris gels (ThermoFisher) and transferred onto nitrocellulose membranes (Bio-Rad). Membranes were blocked for 1 h with 5% milk powder (Campina) or 5% BSA (Sigma) in Tris-buffered saline with TWEEN 20 (TBS-T) and incubated overnight with primary antibodies against ACLY (1:1000, Abcam), phosphorylated ACLY (1:1000, Cell Signaling Technology), and α-Tubulin (1:2000, Sigma-Aldrich, #T5168). Next day, membranes were incubated with horseradish peroxidase (HRP)-conjugated secondary antibodies (1:2000, Thermo Fisher Scientific) in milk or BSA (5% in TBS-T) and developed using SuperSignal West Pico Chemiluminescent PLUS Substrate (Thermo Fisher Scientific).

For histone acetylation analysis, BMDMs were washed with cold PBS and lysed on ice in Triton Extraction Buffer (0.5% Triton X-100, 0.02% (w/v) $NaN_3$ in PBS) supplemented with fresh protease inhibitor cocktail (Sigma-Aldrich). Lysates were scraped, collected and centrifuged for 10 min at 2000 × g and 4 °C. Subsequently, pellets were resuspended in 0.2 M HCl and rotated overnight at 4 °C for acid histone extraction. Next day, samples were centrifuged for 10 min at 2000 × g and 4 °C, and supernatants (containing the extracted histones) were collected. Immunoblotting of histone extracts was performed as detailed above, with primary antibodies directed against H3 (1:5000, Cell Signaling Technology) and H3K27ac (1:1000, Diagenode).

**Gene expression analysis.** Aortic roots were disrupted with glass beads in a MagnaLyser system and total RNA was extracted from homogenates using TRIzol reagent (Life Technologies) and the RNeasy Mini Kit with DNase treatment (QIAGEN). For BMDMs, total RNA was isolated from macrophages in 24-well plates ($5 \times 10^5$ cells/well) using the RNeasy Mini Kit with DNase treatment (QIAGEN) or GeneJET RNA Purification Kit (Thermo Fisher Scientific). Isolated RNA from aortic arches and BMDMs was reverse-transcribed into cDNA using the High-Capacity cDNA Reverse Transcription Kit (ThermoFisher) and gene expression analysis was performed with SYBR Green Fast mix (Applied Biosystems) on a ViiA7 system (Applied Biosystems). Levels of expression were normalized to those of housekeeping genes Rplp0 and Ppia. Primer sequences used for gene expression analysis are provided in Supplementary Table 2.

**Cytokine, NO, and ROS production.** Production of IL-6, TNF, and NO was quantified in supernatants using ELISA (Life Technologies) or the Griess test (Sigma) respectively, following the supplier's protocols. In order to determine ROS production, macrophages were incubated with 20 μM CM-$H_2$DCFDA (Thermo-Fisher) for 30 min at 37 °C (5% $CO_2$). Hereafter, cells were washed with PBS and fluorescence was measured on a Beckman Coulter CytoFLEX.

**Flow cytometry.** To examine surface marker expression, BMDMs were detached using citrate buffer (17 mM tri-Sodium citrate dehydrate and 135 mM potassium chloride in water), transferred to V-bottom well plates, incubated with 1:50 anti-CD16/CD32 Fc-block and stained for 20 min at room temperature in the dark with antibodies. Antibodies were directed to CD40-APC (Biolegend, 1:100), MHC-II-PerCP/Cy5.5 (Biolegend, 1:200), CD80-BV650 (Biolegend, 1:100) and CD86-BV510 (Biolegend, 1:100) for LPS-stimulated macrophages or directed to CD71-PE (BD PharMingen, 1:250), CD206-APC (Biolegend, 1:250), CD273-PE (BD Phar-Mingen, 1:250), CD301-AF647 (Serotec, 1:250) or isotype controls (Biolegend) for IL-4 stimulated macrophages (IGG2A-PE, IGG2A-APC), diluted in staining buffer (PBS with 0.5% BSA and 2.5 mM ethylenediaminetetraacetic acid; EDTA). After staining, cells were washed and resuspended in staining buffer, measured on a Beckman Coulter CytoFLEX or BD LSR Fortessa and analyzed using FlowJo (Tree Star). Surface expression was calculated as ΔMFI = (median fluorescence intensity)$_{positive\ staining}$ − (median fluorescence intensity)$_{isotype\ or\ FMO\ control}$. MFI controls (isotype or FMO) were measured on a pool of all samples.

**Metabolomics.** Macrophages were quenched by ice-cold methanol. $D_3$-aspartic acid, $D_3$-serine, $D_5$-glutamine, $D_3$-glutamate, $^{13}C_3$-pyruvate, $^{13}C_6$-isoleucine, $^{13}C_6$-glucose, $^{13}C_6$-fructose-1,6-biphosphate, $^{13}C_6$-glucose-6-phosphate, adenosine-$^{15}N_5$-monophosphate and guanosine-$^{15}N_5$-monophosphate (5 μM) served as internal standards. After addition of chloroform, samples were centrifuged and the

"polar" top layer was dried in a vacuum concentrator and dissolved in methanol/water (6/4, v/v). Analysis was done using a Thermo Scientific (U)HPLC system coupled to a Thermo Q Exactive (Plus) Orbitrap mass spectrometer (Waltman) with a SeQuand ZIC-cHILIC column. MS data were acquired in full scan, negative ionization mode with a mass resolution of 140,000[46]. Interpretation of the data was performed on the Xcalibur software (ThermoFisher). Further analyses were performed in R using *ggplot2*, *ropls*, and *mixOmics* packages.

**Metabolic determinants and extracellular flux analysis.** Glucose (GOD-PAP method, Biolabo) and ATP (ThermoFisher) levels were enzymatically determined according to manufacturer's protocol. Lactate was enzymatically determined on 3% metaphosphoric acid (MPA) deproteinized cell culture supernatants. After centrifugation ($20,000 \times g$, 10 min) supernatants and L-lactic acid standards were measured for background fluorescence in a 0.5 M Glycine-0.4 M Hydrazine buffer, pH 9.0 with 3.6 mM NAD with a Mithras LB-940 reader at λex/λem = 340–10/450–10 nm five times every 2 min. Reactions were started by adding 0.5 M Glycine-0.4 M Hydrazine buffer, pH 9.0 with 50 µg/mL L-lactate dehydrogenase from rabbit muscle (Roche) and again measured at λex/λem = 340–10/450–10 nm every 2 min until a stable read was achieved.

For extracellular flux analysis, the XF-96 Flux Analyzer (Agilent) was used to assess OCR and ECR on macrophages from both phenotypes. BMDMs ($8 \times 10^4$ cells/well) were plated on XF-96-cell culture plates (Agilent), treated with LPS or left untreated. In all, 1 hr before the assay, cells were washed and replaced with Dulbecco's Modified Eagle's Medium (DMEM) (Sigma-Aldrich) without glucose, phenol red, and sodium bicarbonate. The run consisted of 2 min mixing, 3 min measuring and subsequent four injections: glucose (final assay concentration 25 mM), oligomycin A (OM, final assay concentration 1.5 µM), FCCP (final assay concentration 1.5 µM) with sodium pyruvate (final assay concentration 1 mM), with last injection of antimycin A (AA, final assay concentration 2.5 µM) with rotenone (final assay concentration 1.25 µM). Data were analyzed using Wave software as detailed before[50].

**Transcriptomics.** Total RNA was isolated from BMDMs using a RNeasy Mini Kit with DNase treatment (QIAGEN) and strand-specific libraries were constructed using the KAPA mRNA HyperPrep kit (KAPA Biosystems). Subsequently, samples were sequenced on an HiSeq 4000 instrument (Illumina)[46]. Afterwards, reads were aligned to the mouse genome mm10 by *STAR 2.5.2b*. BAM files were indexed and filtered with *SAMtools* and raw tag counts and RPKM values were summed using HOMER2's analyzeRepeats.pl script. Differential expression was assessed using the *DESeq2* package in an R environment. Reactome pathway enrichment analysis was performed using the *ReactomePA* package for significantly differentially expressed genes (adjusted $p$ value $\leq 0.05$). Significantly enriched pathway list was filtered on redundant pathways that represent parent functionality. Summarized pathway list was visualized using ridge plots, by extracting gene ids and plotting fold change distributions using in-house scripts. Volcano plot and heatmaps were generated using the *ggplot2*, *ggrepel*, and *pheatmap* package.

**Analysis of sterols in macrophages.** Sterols were analyzed in $2 \times 10^6$ macrophages, homogenized in 100 µl PBS followed by sonication in a sonicator bath for 2 min. One hundred microliters of internal standard solution (65 nmol of epico-prostanol) was added and mixed followed by addition of 1.0 ml of saponification reagent (0.3 M KOH in ethanol). This was incubated for 2 h at 80 °C. After cooling, 2 ml of hexane and 0.5 ml of water was added and mixed. Three hundred microliters of the hexane layer was taken to dryness under a flow of nitrogen at 40 °C and 120 µl of BSTFA (+1% TMCS) was added and incubated for 30 min at 80 °C. One microliter of the resulting trimethylsilyl (TMS)-ethers of the sterols was injected on a GC system (CPSil5 column, Agilent GC 7890B) with FID detection for quantification using the added internal standard. The identity of the sterols was confirmed by GC-MS using separation on a Agilent GC 7890B followed by selected ion monitoring of the TMS-derivatives on an MSD5977A MS detector in the EI + -mode.

**Metabololipidomics.** Lipids were extracted from macrophages ($5 \times 10^5$ cells/condition) in 24-well plates, cultured in serum-free phenol red-free DMEM medium supplemented with 25 mM HEPES, 2 mM L-glutamine, 100 U/ml penicillin, and 100 µg/ml streptomycin. Cell lysates were quenched by adding 1.5 mL methanol and 4 µl internal standard solution [2H4]LTB4, [2H8]15-HETE, [2H4]PGE2, [2H5]DHA, [2H11]14(15)EET and [2H4]PGF2α (50 ng/mL in MeOH) was added. Samples were then placed in −20 °C to equilibrate and spun down for 10 min at 16.200 g at 4 °C. Supernatants were diluted in 7.5 mL H$_2$O, pH was corrected to 3.5 using formic acid (99%) and subsequent solid-phase extraction was used to clean the samples further. Samples were loaded on C18 Cartridges (Sep-Pak, Vac3 3cc (200 mg)) prewashed with MeOH and H$_2$O and subsequently washed with H$_2$O and n-hexane. Extracted lipids were eluted using methylformate and collected in glass tubes. The elutes were dried at 40 °C using a N2 flow, reconstituted in 200 µL MeOH (40%) and transferred to deactivated glass inserts.

Oxylipid content of the samples was measured using a targeted HPLC-MS/MS method according to the protocol of Jónasdóttir et al.[51]. Separation of lipids was accomplished using a Shimadzu LC-system, which consisted of two LC-30AD

pumps, a SIL-30AC autosampler (at 4 °C) and a CTO-20AC column oven (50 °C). Samples were injected (40 µL) and compounds were separated on a Kinetex C18 column (50 × 2.1 mm, 1.7 µm column), which was protected with a C8 pre-column (Phenomenex). A gradient of H$_2$O (eluent A) and MeOH (eluent B), both with 0.01% acetic acid, was used to elute the samples at a constant flowrate of 400 µL/min. Other machine settings were in line with the protocol of Jónasdóttir[51] and scheduled MRM mode was used to detect compound peaks. Compounds were detected using their relative retention times together with characteristic mass transitions and these and other individually optimized parameters can be found in the Supplementary table 3.

**Cell cycle analysis.** After 48 h of starvation in medium without LCM, macrophages ($1 \times 10^6$ cells/well) in 12-well plates were washed with PBS and subsequently detached with citrate buffer (17 mM tri-Sodium citrate dehydrate and 135 mM potassium chloride in water) and transferred to tubes. Cells were centrifuged and ice-cold 70% ethanol was drop-wise added to the pellet while vortexing. After fixation at 4 °C for at least 24 h, cells were washed two times with PBS and permeabilized with 0.1% Nonidet P-40 for 10 min. Permeabilized cells were transferred to V-bottom 96-well plates and washed with PBS. Subsequently, cells were resuspended in PBS with 50 µg/ml propidium iodide and 25 µg/ml RNAse and incubated for 30 min at room temperature. After staining, cell cycle status was determined with flow cytometry.

**In situ staining of senescence-associated β-galactosidase activity.** β-Galactosidase activity was assessed in situ after culturing cells on glass coverslips. Cultured cells were fixed with 4% paraformaldehyde for 5 min and incubated overnight at 37 °C in senescence-associated b-galactosidase staining solution (0.1% 5-bromo-4-chloro-3-indolyl-β-D-galactopyranoside, 5 mM ferrocyanide, 5 mM ferricyanide, 150 mM NaCl, 2 mM MgCl, 40 mM citric acid, 40 mM sodium phosphate, pH 6.0). Coverslips were mounted with Mowiol and captured using a bright field microscope using a ×10 objective. Cells were quantified using ImageJ cell counting software.

**In vitro efferocytosis.** A day before the assay, macrophages were stained for 20 min with 2.5 µl/ml DiD-stain, washed with PBS and seeded ($10^5$ cells/well) in 96-well plates. On the day of the assay, RAW264.7 cells were killed by a heat shock of 70 °C for 5 min. Subsequently, killed cells were stained with pHrodo at a concentration of 2 mg/ml for 30 min at RT and washed with PBS. Dead RAW264.7 cells were resuspended in medium and incubated as 5:1 with the DiD-stained macrophages for 1.5 h. After washing, uptake of pHrodo-stained dead cells by DiD-stained macrophages was determined with flow cytometry and validated by imaging flow cytometry (ImageStream). A 4 °C incubation served as control to set the positive pHrodo gate.

**Statistical analysis.** Data are presented as mean ± standard error of the mean (SEM). Statistical significance of data was tested using a two-tailed Student's $t$ test (when comparing two groups), ordinary one-way analysis of variance (ANOVA) followed by Bonferroni's post hoc comparison with test multiple groups or two-way ANOVA followed by Bonferroni's post hoc comparison for grouped analyses in GraphPad Prism software. $P$ values $< 0.05$ were considered significant, with levels of significance being indicated as follows: $*p < 0.05$; $**p < 0.01$; $***p < 0.001$.

**Reporting summary.** Further information on research design is available in the Nature Research Reporting Summary linked to this article.

## Data availability

Any remaining data supporting the results of the study will be made available from the corresponding author upon reasonable request. RNA-sequencing data is deposited in the GEO-database under accession number: GSE126690. Raw data from lipidomics measurements are provided as Supplementary Data 1. Raw data from metabolomics measurements are deposited under accession number MTBLS2159 in the MetaboLights database. Source data are provided with this paper.

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

## Acknowledgements

J.V.d.B. received a VENI grant from ZonMW (91615052) and a Netherlands Heart Foundation junior postdoctoral grant (2013T003) and senior fellowship (2017T048). M.P.J.d.W. is an established investigator of the Netherlands Heart Foundation, is supported by grants from the Netherlands Heart Foundation and Spark-Holding BV (2015B002 and 2019B016), the European Union (ITN grant EPIMAC and REPRO-GRAM [EU Horizon 2020]), and Fondation Leducq (16CVD-01), and holds an AMC fellowship. We acknowledge support from Cancer Center Amsterdam, Amsterdam Cardiovascular Sciences, the Netherlands CardioVascular Research Initiative (CVON GENIUS1 and GENIUS2), Dutch Federation of University Medical Centers, the Netherlands Organisation for Health Research and Development, the Royal Netherlands Academy of Sciences. We would like to thank Roelof Ottenhoff for his efforts in arranging the transfer of mice to our animal facility. Technical assistance with TUNEL staining was received from Jacques Debets. Sterols were analyzed by the Core Facility Metabolomics of the Amsterdam UMC, location AMC. We would like to thank Ron Heeren and Jian-Hua Cao from M4I Division of Imaging Mass Spectrometry of Maastricht University for their efforts in the measurement of lipids in the plaques.

## Author contributions

J.B. and S.G.S.V., and M.P.J.d.W. and J.V.d.B. contributed equally to this work. Conceptualization, J.V.d.B.; methodology, J.B., S.G.S.V.; formal analysis, J.B., S.G.S.V., J.Y.B., K.H.M.P., M.v.W., and S.L.; investigation, J.B., S.G.S.V., S.v.d.V., M.J.J.G., C.P.P.A.v.R., J.C.S., J.Y.B, G.R.G., E.M., A.E.N.; writing—original draft, J.B., S.G.S.V.; writing—review & editing, J.B., S.G.S.V., J.C.S., D.M., G.K., H.E.d.V., E.L., K.E.W., M.P.J.d.W., J.V.d.B.; visualization, J.B., S.G.S.V.; funding acquisition, M.P.J.d.W. and J.V.d.B. All authors read and approved the final manuscript.

## Competing interests

The authors declare no competing interests.
