## [Peer Review File · Nature Communications]

Reviewers' comments:

Reviewer #1 (Remarks to the Author):

Baardman and colleagues have investigated the role of myeloid cell ATP citrate lyase (Acy) in atherosclerosis. The findings related to atherosclerosis are novel and interesting, in part because an Acy inhibitor (Bempedoic acid) is being tested in humans as a new lipid-lowering therapy. The main findings of the present study show that myeloid cell-targeted Acy-deficiency results in a more stable atherosclerotic lesion phenotype, characterized by smaller necrotic cores and thicker fibrous caps. The mechanistic data obtained in isolated macrophages are more complicated, and the mechanism proposed by the in vitro studies is not well supported by the in vivo data. In vitro, isolated Acy-deficient macrophages exhibited deregulated cholesterol biosynthesis, reduced LXR activation, increased susceptibility to apoptosis, and an increased efferocytotic capacity. These processes were not investigated in lesional macrophages. The authors conclude that targeting macrophage Acy is a promising therapeutic target to stabilize atherosclerotic plaques.

Major comments:

1. The lesions are not sufficiently well characterized and the link between the in vitro mechanistic studies and atherosclerosis is underdeveloped. For example, are there differences in macrophage or smooth muscle cell content in lesions from the Acy-deficient chimeras? The explanation that necrotic cores are smaller because Acy-deficient macrophages are more prone to apoptosis but also have an increased ability to efferocytose apoptotic cells is complicated. Is macrophage efferocytosis or apoptosis predominant in lesions (an estimate of efferocytosis in lesions can be obtained by TUNEL-macrophage co-localization (see e.g., PMID:18451332)? The study would be strengthened by isolation and analysis of the phenotype or transcriptome of lesional macrophages.

2. The mechanism whereby reduced acetyl-CoA levels lead to the in vitro phenotypes in macrophages is not sufficiently well described. Are ATP levels or mitochondrial function altered by Acy-deficiency? Are Acy-deficient macrophages more lipid loaded in vivo or in vitro, given the increased expression of proteins involved in lipid uptake and the reduced expression of cholesterol exporters? The reduced expression of Abca1 would support a lipid-loaded phenotype.

3. Myeloid cell-targeted Acy-deficiency reduced plasma triglycerides. This interesting phenotype should be described in greater detail. Was hepatic triglyceride production increased? Could the reduced plasma triglycerides contribute to the effect of Acy-deficiency in lesions?

Additional comments:

4. The human lesion data need to be quantified. How many lesions were examined? The human subjects and tissue collection should be described. The data would also be strengthened by the use of a second macrophage marker because CD68 can be expressed by smooth muscle cells. It would be informative to have similar data on p-AcLy localization in the mouse lesions.

5. Macrophage markers (mRNA) should be measured in the aortic arches. Without this information it is difficult to know to what extent the increased inflammatory phenotype is due to the presence of more macrophages in the Acly-deficient animals. In addition, was the increased Tgfb1 due to increased expression in macrophages? Did the authors measure Tgfb1 mRNA in vitro?

6. The meaning of the reduction in blood B-cells is not discussed.

7. There are some technical issues. The number of independent experiments performed is not always clear. Several of the legends state "Values represent mean +/- SEM (n=3 technical replicates of one representative experiment)." How many independent experiments were performed to support the conclusions? How many replicates were used for the RNA-seq experiments?

8. Acly phosphorylation has previously been shown to be increased by IL-4 (PMID:26894960). How are these findings consistent with the present study, which shows increased Acly phosphorylation in response to LPS?

Reviewer #2 (Remarks to the Author):

In this manuscript, the authors describe a novel discovery that myeloid ATP citrate lyase (Acly) deficiency induces a stable plaque phenotype characterized by increased collagen deposition and fibrous ca thickness, along with a smaller necrotic core by using conditional genetic knockout mouse model. In general, this is a good observational paper. The discovery is interesting. However, several important results are missing. ATP citrate lyase is the key enzyme to generate acetyl CoA in cytosol. Acetyl CoA is the substrate for synthesis of cholesterol, free fatty acids, and may serve as epigenetic

modulator for histone acetylation in nuclei as the authors proposed. The mechanism by which the knockout stabilize atherosclerotic plaques is unclear.

The major comments

1. It will be helpful to show cholesterol, fatty acids, and histone acetylation levels following the genetic knockout of Acly in macrophages in vitro and in vivo. The mRNA levels cannot explain its function because there is a feedback mechanism.

2. Knockout of Acly, theoretically, will decrease or block cholesterol and free fatty acid biosynthesis, which will induce cell death and make the atherosclerosis worse.

3. Inhibitors of acetyl CoA carboxylase (ACC) has been used in clinical trials. The enzyme is different from Acly. ACC is the first key enzyme for biosynthesis of cholesterol and free fat acids. Acly is to provide acetyl CoA but acetyl CoA can also be provided by pyruvate oxidation from glycolysis, which is the major source of acetyl CoA. Therefore, its mechanism has a big question. Histone acetylation data will help a lot.

Reviewer #3 (Remarks to the Author):

The manuscript by Baardman et al reports some interesting data on the potential role of the enzyme ATP citrate lyase (ACLY) in macrophage phenotype and inflammatory function. Specifically, they show that ACLY is phosphorylated within inflammatory macrophages and in atherosclerotic plaques, and, furthermore, that ACLY deletion in myeloid cells alters the phenotype of these plaques, albeit not macrophage inflammatory function in vitro. Mechanistically, they propose this is due to alterations in macrophage cholesterol metabolism and also to increased apoptosis and efferocytosis of ACLY-deficient macrophages. These data are interesting, novel and potentially of significance for clinical translation and identification of therapeutic targets, however I feel that the data as they are presented don't fully support the conclusions drawn, and additionally that some important mechanistic insight is lacking.

Significant concerns:

1. The authors genetically manipulate ACLY expression in macrophages, because they show this enzyme to be phosphorylated (and therefore presumably more active) in both inflammatory macrophages in vitro, and also in atherosclerotic plaques in vivo (but see minor concern below). However, they do not confirm increased activity of ACLY in inflammatory macrophages, by either measuring enzyme activity or by analysis of the abundance of the substrate (citrate) or product (acetyl-coA) of this enzyme, which would be important supportive data for their hypothesis. They also do not perform any metabolic analysis of ACLY-deficient macrophages to confirm that their genetic manipulation results in the expected metabolic changes, and assess for any other metabolic implications of ACLY deletion. Indeed, no metabolic analyses are performed in this manuscript, which would provide significant mechanistic insight into how this metabolic enzyme impacts macrophage phenotype and development of atherosclerotic plaques.

2. The authors have performed in-depth transcriptional analyses (RNA-seq) of WT and ACLY-deficient macrophages. These data highlight some interesting differences between the two cell types, for example in cholesterol and fatty acid metabolism, however without any accompanying metabolic analysis it is challenging to mechanistically link ACLY deficiency with the transcriptional changes observed. For example, the authors speculate that the changes in cholesterol metabolism are caused by reduced availability of acetyl-coA (as a compensatory mechanism), but they have not confirmed this. Even if (as expected) acetyl-CoA is reduced in abundance, to formally link this to altered cholesterol metabolism, additional experiments using an “add-back” approach or an alternative, pharmacological approach to modulate acetyl-CoA abundance would be very helpful.

3. The authors also speculate that decreased LXR activity, secondary to reduced abundance of the LXR-activator desmosterol, could explain the elevated inflammatory responses and increased apoptosis of ACLY-deficient macrophages. Again, this could be formally tested by adding back desmosterol and assessing inflammatory macrophage phenotype and function, as well as cell cycle progression and apoptosis, but this experiment and the potential key mechanistic insight it could provide is lacking.

4. In figure 1l the authors show that TGF β 1 transcripts are increased in the aortic arch of ACLY-MKO mice, with the inference that this cytokine is having a pro-fibrotic, anti-atherogenic role. To prove this, it would be important to measure not only mRNA but also (active) TGF- β protein and to formally test its role by neutralisation/signalling blockade in vivo, using specific antibodies or pharmacological inhibitors.

Minor concerns:

1. The legend for Figure 1a doesn't accurately describe the data presented – mRNA data as well as protein data are shown but only the protein/WB data are described.

2. The resolution and scale of the image presented in Figure 1b make it very difficult to discern if p-ACLY and CD68 are co-localised. Can the authors improve this? Also it may be helpful to have some quantitation of co-localisation in the image. Furthermore, total ACLY staining (i.e. phosphorylated and non-phosphorylated) is an important control which is lacking here.

3. ACLY abundance in WT vs KO is only shown in macrophages. It would be helpful to see the level of knockdown in other cells types, particularly other myeloid cells.

4. In the legend of figure 2 the statement “Representative histograms and quantified surface expression ($\Delta\text{MFI} = [\text{Median fluorescence intensity}]_{\text{positive staining}} - [\text{MFI}]_{\text{control}}$) are shown” is given but it is unclear whether the “control” is the WT or the paired WT or KO – please can the authors clarify this. It would be most appropriate to use the paired WT or KO respectively.

Reviewer #1 (Remarks to the Author):

Baardman and colleagues have investigated the role of myeloid cell ATP citrate lyase (Acy) in atherosclerosis. The findings related to atherosclerosis are novel and interesting, in part because an Acy inhibitor (Bempedoic acid) is being tested in humans as a new lipid-lowering therapy. The main findings of the present study show that myeloid cell-targeted Acy-deficiency results in a more stable atherosclerotic lesion phenotype, characterized by smaller necrotic cores and thicker fibrous caps. The mechanistic data obtained in isolated macrophages are more complicated, and the mechanism proposed by the in vitro studies is not well supported by the in vivo data. In vitro, isolated Acy-deficient macrophages exhibited deregulated cholesterol biosynthesis, reduced LXR activation, increased susceptibility to apoptosis, and an increased efferocytotic capacity. These processes were not investigated in lesional macrophages. The authors conclude that targeting macrophage Acy is a promising therapeutic target to stabilize atherosclerotic plaques.

Major comments:

1. The lesions are not sufficiently well characterized and the link between the in vitro mechanistic studies and atherosclerosis is underdeveloped. For example, are there differences in macrophage or smooth muscle cell content in lesions from the Acy-deficient chimeras? The explanation that necrotic cores are smaller because Acy-deficient macrophages are more prone to apoptosis but also have an increased ability to efferocytose apoptotic cells is complicated. Is macrophage efferocytosis or apoptosis predominant in lesions (an estimate of efferocytosis in lesions can be obtained by TUNEL-macrophage co-localization (see e.g., PMID:18451332)? The study would be strengthened by isolation and analysis of the phenotype or transcriptome of lesional macrophages.

We agree with the reviewer that the mechanistic data obtained in vitro are complicated. Based on this comment and others, we further studied the proposed mechanisms and adapted the manuscript accordingly. Moreover, the revised version of this manuscript now links in vitro and in vivo better as suggested by the reviewer as specified below:

To assess differences in macrophage and neutrophil counts, we performed MOMA-2 and Ly6G staining in aortic roots of control and Acy^{M-KO}-transplanted mice. Quantifying macrophage and neutrophil content revealed no differences in macrophage and neutrophil numbers in the control versus the Acy^{M-KO}-transplanted groups, shown in figure 1n,o.

Fig 1. (n) Quantification of MOMA-2⁺ area for macrophages. **(o)** Quantification of Ly6G⁺ area for neutrophils

Moreover, assessing *Cd68* gene expression in aortic arches also did not reveal differences between both groups, together indicating that the distinct (inflammatory) plaque phenotype probably cannot be explained by differences in macrophage content. *Cd68* gene expression data are shown in supplementary figure 2c.

Supplementary Fig 2. (c) *Cd68* expression in aortic arches

Measuring smooth muscle cell (SMC) content based on the use of antibodies against for example smooth muscle actin (SMA) is challenging in mouse plaques since SMC can switch phenotypes, lose specific SMC markers expression like MYH11, and acquire macrophage-like characteristics, including expression of CD68 and decreased expression of SMC specific markers¹⁻⁴. Therefore, we also functionally assessed the fibrotic activity of SMCs by measuring collagen content as a commonly used functional readout. As detailed in figure 1k-m, plaques from *Acly*^{M-KO}-transplanted mice showed increased collagen content as well as to increased *Tgfb1* expression in those plaques (Figure 1p). Together these data indicate increased pro-fibrotic smooth muscle cell activity.

Fig 1. (k) Sirius red staining of plaques. Scale bar represents 200 μm . **(l)** Quantification of the collagen deposition. **(m)** Minimal cap thickness was measured at the thinnest region of the fibrotic cap surrounding the necrotic core as indicated by the arrows in panel (k). **(p)** Normalized expression of *Tgfb1* in aortic arches of control and *Acly*^{M-KO} mice.

We set out to detect apoptosis in the plaques of control versus *Acly*^{M-KO}-transplanted mice by TUNEL staining. Plaques of the *Acly*-deficient group showed increased numbers of TUNEL⁺ apoptotic cells per μm^2 . These new *in vivo* data are now shown in Figure 5e and further support our *in vitro* data in which we observed increased apoptosis in *Acly*-deficient macrophages (Figure 5b,c).

Measuring TUNEL/macrophage colocalization as an estimate of efferocytosis is a great suggestion of the reviewer. We did try this according to the protocol detailed in the reference⁵. *Acly* deficient lesions show a tendency for an increased ratio of bound TUNEL⁺ cells to free TUNEL⁺ cells, indicating that *in vivo* efferocytosis follows the same trend as *in vitro* cultured macrophages.

This has been addressed in the manuscript as follows:

To validate these results in vivo, we analyzed apoptosis and efferocytosis in the atherosclerotic plaques from control and AclyM-KO mice by terminal deoxynucleotidyl transferase dUTP nick end labeling (TUNEL) double staining with MAC3. In correspondence

with our *in vitro* data, apoptosis and to a lesser extent efferocytosis appeared to be increased in plaques from *Acly*^{M-KO} mice (Fig. 5e,f).

We agree with the reviewer that it would be interesting to isolate and phenotype lesional macrophages in control vs *Acly*^{M-KO} mice. In our opinion, single-cell RNA-seq analysis would be needed to gather insightful data concerning the transcriptome of macrophages in control versus *Acly*^{M-KO} plaques. These are very extensive and expensive analyses and recently indicated a complexity within the myeloid cell compartment that cannot be mapped by bulk analysis of isolated lesional macrophages^{6,7}. Such in-depth analysis of plaque macrophages is something we envision to do in the support of future grants but meanwhile we still wanted to further evaluate macrophages closely related to lesional macrophages. To validate the increase in inflammatory response that we observed in *Acly*-deficient BMDMs *in vitro*, we now also performed *ex vivo* analysis of peritoneal macrophages isolated from control and *Acly*^{M-KO} atherosclerotic mice. This is an *in vivo* model of macrophage foam cells that is commonly used in the field to relate *in vitro* observations to the *in vivo* context⁸⁻¹¹. As shown in the new supplementary figure 2e, peritoneal macrophages from control and *Acly*^{M-KO} atherosclerotic mice show a similar trend as *in vitro* cultured BMDMs in increased expression of pro-inflammatory genes (Supplementary Fig. 2e).

This was included in the text as follows:

Likewise, we detected increased inflammatory gene expression *in vivo* in the atherosclerotic lesions and peritoneal macrophages of the *Acly*^{M-KO} group (Fig. 2e, Supplementary Fig. 2e).

Supplementary Fig 2. (e) Inflammatory gene expression in untreated or 24 hour LPS treated peritoneal cells from control and *Acly*^{M-KO} *LDLr*^{-/-} mice.

2. The mechanism whereby reduced acetyl-CoA levels lead to the *in vitro* phenotypes in macrophages is not sufficiently well described. Are ATP levels or mitochondrial function altered by *Acly*-deficiency? Are *Acly*-deficient macrophages more lipid loaded *in vivo* or *in vitro*, given the increased expression of proteins involved in lipid uptake and the reduced expression of cholesterol exporters? The reduced expression of *Abca1* would support a lipid-loaded phenotype.

To better define the mechanism whereby *Acly* deletion leads to the observed *in vitro* phenotypes in macrophages, we performed a broad range of new analysis:

Firstly, we performed metabolomics analysis that revealed that *Acly* deletion in macrophages did not reduce acetyl-CoA levels. It should be noted that acetyl-CoA levels were always low or even undetected in our samples, and are therefore not shown in the manuscript. Nevertheless, this indicates that macrophages can rescue *Acly*-mediated acetyl-CoA production (presumably via acetate and *Acss2*, see later). Importantly, this new analysis did reveal increased LPS-induced citrate levels in *Acly*-deficient macrophages, which is expected since citrate is normally converted into acetyl-CoA by *Acly*. Other TCA cycle intermediates however, remained unaltered as now shown in Supplementary Figure 3b.

To profile mitochondrial function and ATP production in more detail, we performed extracellular flux (Seahorse) analysis in control vs *Acly*-deficient macrophages. These analyses revealed a similar LPS-induced metabolic switch in control and *Acly*-deficient macrophages as exemplified by comparable glycolysis parameters, OXPHOS parameters and ATP production rates via both pathways. In line with this, glucose usage and lactate production were identical in control and *Acly*^{M-KO} macrophages as measured in supernatants and also total ATP levels appeared similar. Together, these data reveal no major defects in the assessed core metabolic pathways and this notion is supported by the similar energy charge calculated based on AMP, ADP and ATP levels in control and KO cells. Mitochondrial analysis by seahorse, glycolysis parameters, cellular ATP concentrations and energy charge data is now presented in Supplementary Figure 3c-g.

It is in described in the manuscript as follows:

*To identify the mechanism by which myeloid deletion of *Acly* influences macrophage and plaque phenotype, we explored changes in core metabolic pathways as a potential explanation of the increased inflammatory response in *Acly*-deficient macrophages.*

*Metabolomics revealed that the expected LPS-induced accumulation of citrate was more pronounced in the absence of *Acly*, signifying decreased *Acly*-mediated conversion of citrate into acetyl-CoA^{19,20}. Yet, loss of *Acly* led to no further changes in the abundance of other TCA cycle intermediates (supplementary Fig. 3b). Extracellular flux analysis, and glucose, lactate and ATP measurements showed no differences in mitochondrial function, glycolysis and ATP production, indicating that other mechanisms should explain the observed inflammatory phenotypes of *Acly*-deficient macrophages (supplementary Fig. 3c-g).*

Supplementary Fig 3. Mitochondrial function remains unaltered in *Acly*^{M-KO} BMDMs (a) Differentiated control and *Acly*^{M-KO} macrophages were left untreated or stimulated with LPS (b) relative abundance of TCA cycle intermediates (c) Lactate secretion after 24 hours LPS treatment (d) Metabolized glucose in 24 hours (e) Cellular energy charge as calculated by $[ATP]+0.5[ADP]/([AMP]+[ADP]+[ATP])$ (f) Cellular ATP concentration (g) Seahorse analyses and extracted parameters. Values represent mean \pm SEM (n=6 technical replicates from 3 pooled mice) ***P < 0.001 by two-way ANOVA with Bonferroni post hoc test for multiple comparisons

As also stated by the reviewer, the increased expression of proteins involved in lipid uptake and the reduced expression of cholesterol exporters might hint towards a lipid-laden phenotype of *Acly*^{M-KO} macrophages. We now investigated this using an extensive set of assays. Probing free cholesterol by Filipin or neutral lipids by Bodipy 493/503 staining, followed by flow cytometric analysis indicated no differences in free cholesterol and neutral lipid levels between both genotypes. Both control and *Acly*^{M-KO} BMDMs did not show clear signs of lipid loading as demonstrated by the absence of Oil red O stain in both groups and total lipid abundance as measured by metabololipidomics is also similar. Measuring total cholesterol by gas chromatography-mass spectrometry (GC-MS) showed increased total cholesterol levels in *Acly* deficient macrophages, which indicates that they manage to restore cholesterol levels. These new data are now indicated in figure 3g and figure 4a,b.

In our view, *Acly* deletion forces the macrophages to rewire their metabolism to secure cholesterol metabolism. To do so, the cells maximize uptake and minimize export of cholesterol, resulting in a new equilibrium with minimal metabolic defects but clear phenotypic consequences. This rationale is discussed in the manuscript as follows:

*Apart from cholesterol metabolism, REACTOME pathway analysis also indicated fatty acyl-CoA synthesis and fatty acid metabolism as top deregulated pathways (Fig. 3b). Since Acly provides acetyl-CoA as a precursor for fatty acid synthesis^{22,23}, we measured fatty acid levels in control and *Acly*^{M-KO} macrophages by using lipidomics. Whereas RNA-sequencing analysis suggested a lipid-laden phenotype of *Acly*^{M-KO} macrophages, total lipid pools were similar in control and *Acly*^{M-KO} macrophages (Fig. 4a,b).*

and

*By increasing cholesterol synthesis and import, and via limiting its export, *Acly*-deficient macrophages possibly try to cope with the reduced supply of *Acly*-mediated acetyl-CoA, and by doing so they manage to secure total cholesterol levels (Fig. 3g).*

Fig. 1. Levels of free cholesterol in control and *Acly* deficient macrophages analysed by Fillipin. **Fig. 4 (a)** Total sum of measured lipids by metabololipidomics. **Fig. 4 (b)** Neutral lipids analyzed by Bodipy 493/503. **Fig. 3 (g)** Cholesterol levels in control and *Acly* deficient macrophages.

3. Myeloid cell-targeted *Acy*-deficiency reduced plasma triglycerides. This interesting phenotype should be described in greater detail. Was hepatic triglyceride production increased? Could the reduced plasma triglycerides contribute to the effect of *Acy*-deficiency in lesions?

In the manuscript we described similar plasma cholesterol levels and a reduction in measured plasma triglycerides in the *Acy*^{M-KO} mice. To describe this in more detail, we first measured triglyceride levels in livers. The reduction measured in plasma triglycerides was not accompanied by altered hepatic triglyceride levels, suggesting that its production remains unaffected by myeloid *Acy* deletion. This new data is now shown in supplementary figure 1j.

Supplementary Fig 1. (j) Hepatic triglyceride levels after 10 weeks of HFD

Next, we assessed whether the decreased plasma triglyceride levels in the *Acy*^{M-KO} mice contributed to the plaque phenotype as questioned by the reviewer. To do so, we plotted plasma triglyceride levels to the different plaque phenotypic characteristics. Plasma triglyceride levels did not correlate with collagen content, necrotic core area and plaque size (Supplementary Fig 2d). These data indicate that the observed changes in plasma triglyceride levels did not contribute to the effect we reported after myeloid *Acy* deficiency in lesions.

This supports the earlier notion that plasma cholesterol, but not triglyceride levels positively correlate with plaque size¹²⁻¹⁵. Supporting this idea, cholesterol levels positively correlated with plaque area and necrotic core area in both control and *Acy*^{M-KO} mice. This new information is now included in the revised version of this manuscript as follows:

Plasma triglyceride levels did not correlate with the plaque characteristics, indicating that lower triglyceride levels were not the cause of the more stable plaque phenotype (supplementary Fig. 2d).

Fig Reb.2

Supplementary Fig 2. (d) Correlations of plasma triglycerides with plaque phenotypes. **Fig Reb. 2** Correlation of plasma cholesterol with plaque phenotypes. * $p < 0.05$

Additional comments:

4. The human lesion data need to be quantified. How many lesions were examined? The human subjects and tissue collection should be described. The data would also be strengthened by the use of a second macrophage marker because CD68 can be expressed by smooth muscle cells. It would be informative to have similar data on p-Acyl localization in the mouse lesions.

We now quantified the human lesions as suggested by the reviewer. Hereto, we examined 16 plaques from 10 donors, scoring either as stable (PIT) or as unstable (IPH), which is now described in the methods as follows:

Human plaque tissue collection of the Maastricht Pathology Tissue Collection (MPTC) was in line with the Dutch Code for Proper Secondary use of Human Tissue and respective local Medical Ethical Committees. Formalin-Fixed Paraffin-Embedded (FFPE) human carotid samples collected at autopsy (n=10, mean age 72 years, 64% men) were used for immunohistochemistry. Samples represented the following stages of atherosclerosis (n=5 per group): pathological intimal thickening (PIT) and plaque with intraplaque hemorrhage (IPH) ³⁹.

Phosphorylated Acly predominantly co-localized with CD68⁺ cells in all lesions (Fig. 1c). To exclude a potential contribution of CD68⁺ smooth muscle cells (SMCs), we also assessed co-localization of p-Acyl with CD45 as a general marker for immune cells that, unlike CD68, is not expressed by SMCs¹. Since p-Acyl solely co-localized with CD45⁺ cells in these human atherosclerotic plaques, we

conclude that Acly is mainly activated in CD68⁺ macrophages and not in potential CD68⁺ SMCs with a macrophage-like phenotype (Fig. 1c, supplementary Fig. 1a,b). We next also quantified the total p-Acly⁺ area in all lesions and observed increased p-Acly levels in unstable plaques.

Whereas total Acly was present in most cells in and around the plaque, activated (p-)Acly predominantly co-localized with CD45⁺ CD68⁺ macrophages in human atherosclerotic plaques (Fig. 1b,c, Supplementary figure 1a-c). Additionally, unstable plaques showed increased abundance of activated Acly when compared to stable plaques (Fig. 1d). Rupture-prone plaque areas are known to be dominated by inflammatory macrophages¹³ and together with our observation that the levels of activated p-Acly were increased in inflammatory macrophages, this prompted us to study whether targeting macrophage Acly could improve atherosclerosis outcome.

Fig 1. (b) Immunohistochemical staining for macrophages (CD68) and p-ACLY in human plaques. Scale bar represents 100 μ m. (c) Quantification of co-localization, percentage of p-Acly⁺ macrophages overlapping with CD68⁺ area. (d) Quantification of p-Acly⁺ area in the lesion.

Supplementary Fig 1. (a) Immunohistochemical staining for CD45 and p-Acly (b) colocalization of CD45 and p-Acly

Next, we aimed to localise Acly in the mouse lesions from our study. Since fluorescent staining of p-Acly was not compatible with mouse tissue, we stained for total Acly in lesions from control and Acly^{M-KO} mice. This immunohistochemical analysis demonstrated a reduced expression in macrophage-rich areas in lesions of Acly^{M-KO} mice. Additionally, surrounding tissue that is typically low in macrophages was left unaltered. This new analysis is now shown in supplementary Fig. 2a. Together with our finding that LPS-activated macrophages show increased levels of Acly phosphorylation, but not total Acly (by western blot figure 1a), we conclude that activation of Acly plays a role in inflammatory activation of mouse macrophages. We now discussed this as follows:

After ten weeks of HFD, immunohistochemical and gene expression analysis on lesions confirmed Acly knockdown in myeloid cells in the atherosclerotic plaques of Acly^{M-KO} mice (Supplementary Fig. 2a,b).

Supplementary Fig 2. (a) Immunohistochemical staining of total Acly on lesions from.

5. Macrophage markers (mRNA) should be measured in the aortic arches. Without this information it is difficult to know to what extent the increased inflammatory phenotype is due to the presence of more macrophages in the Acly-deficient animals. In addition, was the increased *Tgfb1* due to increased expression in macrophages? Did the authors measure *Tgfb1* mRNA in vitro?

We addressed this question partly in our answers to question 1 from the reviewer as follows: To assess differences in macrophage and neutrophil counts, we performed MOMA-2 and Ly6G staining in control and Acly^{M-KO}-transplanted mice. Quantifying macrophage and neutrophil content revealed no differences in macrophage and neutrophil numbers in the control versus the Acly^{M-KO}-transplanted group, shown in figure 1n,o. Moreover, assessing *Cd68* gene expression in aortic roots indicated no difference between both groups, together indicating that the distinct plaque phenotypes cannot be explained by differences in macrophage content. *Cd68* gene expression data are shown in supplementary figure 2c.

Fig 1. (n) Quantification of MOMA-2⁺ area for macrophages. (o) Quantification of Ly6G⁺ area for neutrophils **Supplementary Fig 2.** (c) *Cd68* expression in aortic arches

While we could not formally prove that the increased *Tgfb1* transcript levels is due to increased expression in macrophages, staining for TGF-β protein in the lesions revealed that TGF-β co-localized with the presence of macrophages. Whereas TGF-β covers a similar area of the lesion in both genotypes, there is a slight tendency in the Acly^{M-KO} plaque of TGF-β to increasingly co-localize with macrophages. This indicates that increased expression of *Tgfb1* is most likely due to increased expression by macrophages within the lesion.

This is discussed in the manuscript as follows:

*Since labeling of TGF- β and the macrophage marker MOMA-2 suggested an increase of macrophage TGF- β but not total TGF- β (Fig. 1q-s), the increased *Tgfb1* levels were probably caused by macrophages within the lesion.*

As suggested by the reviewer, we did try to detect *Tgfb1* mRNA levels *in vitro* in BMDMs but this experimental setting did not replicate the *in vivo* microenvironment. Indeed, we did not detect *Tgfb1* transcripts in naïve BMDMs, and also not when we exposed the macrophages to LPS as an inflammatory stimulus or to apoptotic cells.

Fig 1. (q) Immunohistochemical staining for macrophages (MOMA-2) and TGF- β in mouse plaques from control or Acly^{M-KO} mice. Scale bar represents 100 μ m. (r) Quantification of co-localization, percentage of TGF- β^+ area in MOMA-2⁺ area. (s) Quantification of TGF- β^+ area in the lesion.

6. The meaning of the reduction in blood B cells is not discussed.

This is a valid remark as our earlier FACS analysis of the blood revealed a reduction of B cells after bone marrow transplantation from control and Acly^{M-KO} mice. We now also analysed B cell levels in the spleen but did not observe differences there (Supplementary figure 1n).

Supplementary Fig 1. (o) Spleen leukocyte composition after 10 weeks of HFD.

Since B cells are from the lymphoid lineage and Acly deletion is driven by the myeloid-specific LysM promoter in our setting, the observed reduction in blood B cells is likely caused by an indirect effect. This is now discussed in the manuscript as follows:

Since splenic B cell levels remained unaltered, there is no apparent general defect in B cell development or activation. It rather suggests an indirect, and yet unknown, effect of myeloid Acly knockdown on circulating B cells.

7. There are some technical issues. The number of independent experiments performed is not always clear. Several of the legends state "Values represent mean +/- SEM (n=3 technical replicates of one representative experiment)." How many independent experiments were performed to support the conclusions? How many replicates were used for the RNA-seq experiments?

To clarify this, we now adapted the figure legends to include this crucial information as highlighted in the revised version of our manuscript.

8. Acly phosphorylation has previously been shown to be increased by IL-4 (PMID:26894960). How are these findings consistent with the present study, which shows increased Acly phosphorylation in response to LPS?

The IL-4-induced Acly phosphorylation described by Covarrubias et al. and our observation that also LPS activates Acly, together demonstrate that different stimuli can activate Acly. Also, this indicates that Acly is involved in different modes of macrophage activation. In fact, the LPS-induced Acly activation that we observed is supported by two recent publications who also detected increased p-Acly in response to LPS (Lauterbach et al. 2019¹⁶, Langston et al. 2019¹⁷). This point is now addressed in the manuscript as indicated below, and is also discussed in more detail in response to question 1 of reviewer 2:

Lauterbach et al. recently applied Acly inhibitors and siRNA-mediated knock down to demonstrate the role of Acly in regulating acute LPS responses via histone acetylation³⁷. The difference between our observations and the ones of Lauterbach et al. might be explained by the knockdown method (i.e. short-term siRNA vs long-term LysM-cre-driven) and other experimental variables. Clearly, the role of Acly in macrophages is stimulus- and time-dependent and also appears to depend on the experimental setting as Namgaladze et al. observed no effect of Acly silencing³⁸. As such, this is an area that warrants future research.

Reviewer #2 (Remarks to the Author):

In this manuscript, the authors describe a novel discovery that myeloid ATP citrate lyase (Acy) deficiency induces a stable plaque phenotype characterized by increased collagen deposition and fibrous cap thickness, along with a smaller necrotic core by using conditional genetic knockout mouse model. In general, this is a good observational paper. The discovery is interesting. However, several important results are missing. ATP citrate lyase is the key enzyme to generate acetyl-CoA in cytosol. Acetyl-CoA is the substrate for synthesis of cholesterol, free fatty acids, and may serve as epigenetic modulator for histone acetylation in nuclei as the authors proposed. The mechanism by which the knockout stabilize atherosclerotic plaques is unclear.

The major comments

1. ***It will be helpful to show cholesterol, fatty acids, and histone acetylation levels following the genetic knockout of Acy in macrophages in vitro and in vivo. The mRNA levels cannot explain its function because there is a feedback mechanism.***

Acetyl-CoA can indeed have different fates and we agree with the reviewer that mRNA levels alone do not give sufficient insight in function. Therefore, we now measured cholesterol, fatty acids and histone acetylation as requested by the reviewer to further support our hypotheses and findings:

First, we evaluated histone acetylation by isolating histones from naïve, LPS- and IL4-treated control and Acy^{M-KO} macrophages and subsequent western blotting with antibodies against histone 3 lysine 27 acetylation (H3K27ac). Confirming previous findings¹⁸, histone acetylation levels were blunted in Acy-deficient IL-4-treated macrophages and this supports the earlier notion that Acy regulates IL-4-induced macrophage activation via histone acetylation. Conversely, Acy deficiency did not affect histone acetylation levels in naïve and LPS-stimulated macrophages.

This new figure has been added to the manuscript in supplemental figure 2e and is shortly discussed as follows:

This decreased IL-4 response may be explained by lower histone 3 lysine 27 (H3K27) acetylation in the absence of Acy and confirms a previous study that applied Acy inhibitors during IL-4-responses in macrophages to link Acy-mediated production of acetyl-CoA to histone acetylation¹⁷. Conversely, H3K27 levels were similar in naïve and LPS-treated control and Acy^{M-KO} macrophages and thus histone acetylation is not the mechanism explaining the macrophage phenotype in this setting.

Supplementary Fig 2. (f) histone 3 K 27 acetylation (H3K27ac) in control and Acy^{M-KO} medium, LPS or IL-4 treated macrophages.

Importantly, Lauterbach et al. recently applied Acly inhibitors and siRNA-mediated knock down to demonstrate the role of Acly in regulating acute LPS responses via histone acetylation¹⁶. This apparent contradiction between our model and the observations of Lauterbach et al. might be explained by the knockdown method (i.e. short-term siRNA vs long-term LysMcre-driven). Clearly, the role of Acly in macrophages is stimulus- and time-dependent and also appears to depend on the experimental setting as Namgaladze et al. observed no effect of Acly silencing¹⁹. This is an area that warrants future research in the field and we are in contact with Eicke Latz (Lauterbach et al. paper) to conduct follow-up studies on this point in the future. Importantly, they also detected increased LPS-induced inflammatory gene expression when investigating their independently generated LysMCre * Acly fl/fl mouse model (personal communication, unpublished data). As such, we are confident that the increased LPS-response in Acly^{M-KO} macrophages is real and we therefore set out to gain further mechanistic insight into this phenotype as detailed below.

Next, we assessed the effect of Acly deficiency on cholesterol levels as suggested by the reviewer. Staining free cytosolic cholesterol with Filipin and neutral lipids with Bodipy 493/503 and measuring total lipid levels by metabololipidomics revealed no differences between control and Acly^{M-KO} macrophages. However, measuring total cholesterol levels revealed that Acly-deficient macrophages maintain their cholesterol levels. Indeed, the deregulated cholesterol metabolism observed in our RNA-sequencing data indicated that macrophages compensate for the decrease in Acly-mediated acetyl-CoA production by increasing the expression of genes involved in cholesterol import and decreasing expression of cholesterol efflux genes. Together, this secures cholesterol biosynthesis and homeostasis, but also affects inflammatory signalling.

Lipid and cholesterol measurements are now added to the manuscript in figure 4a,b and figure 3g respectively and discussed as follows:

Apart from cholesterol metabolism, REACTOME pathway analysis also indicated fatty acyl-CoA synthesis and fatty acid metabolism as top deregulated pathways (Fig. 3b). Since Acly provides acetyl-CoA as a precursor for fatty acid synthesis^{22,23}, we measured fatty acid levels in control and Acly^{M-KO} macrophages by using lipidomics. Whereas RNA-sequencing analysis suggested a lipid-laden phenotype of Acly^{M-KO} macrophages, total lipid pools were similar in control and Acly^{M-KO} macrophages (Fig. 4a,b).

and

By increasing cholesterol synthesis and import, and via limiting its export, Acly-deficient macrophages possibly try to cope with the reduced supply of Acly-mediated acetyl-CoA, and by doing so they manage to secure total cholesterol levels (Fig. 3g).

Fig Reb 1. Levels of free cholesterol in control and *Acly* deficient macrophages analysed by Filipin. **Fig. 4 (a)** Total sum of measured lipids by metabololipidomics. **Fig. 4 (b)** Neutral lipids analyzed by Bodipy 493/503. **Fig. 3 (g)** Cholesterol levels in control and *Acly* deficient macrophages.

Finally, we assessed fatty acids as a third possible fate of *Acly*-derived acetyl-CoA. To provide more insight into the regulation of fatty acid metabolism in the absence of *Acly*, we measured fatty acids and downstream lipid mediators by metabololipidomics. Despite similar levels of total lipids, a slight increase was observed in ω 3 and 6 fatty acids including arachidonic acid (AA), docosahexaenoic acid (DHA), eicosapentaenoic acid (EPA), docosatetraenoic acid/adrenic acid (AdA) and linoleic acid (LA) in *Acly*^{M-KO} (Fig. 4c). RNA-sequencing data revealed a deregulation of the genes (*Ptgs1* and *Ptgs2*) encoding COX-1 and COX-2 enzymes that convert AA into prostanoids such as prostaglandins that are known to influence inflammation (Fig. 4d). Consequently, prostaglandins displayed increased LPS-induced levels in *Acly*^{M-KO} macrophages. Moreover, RNA-sequencing data showed that prostaglandin E receptor 4 (*Ptger4* or EP4) was decreased in *Acly*^{M-KO} BMDMs (Fig. 4e). Since loss of EP4 has been shown to increase inflammatory responses and the PGE₂-EP4 axis is involved in the regulation of inflammation, we pre-treated control and *Acly*^{M-KO} LPS-activated BMDMs with PGE₂. Indeed *Acly*^{M-KO} macrophages display a reduced responsiveness to PGE₂ as compared to control macrophages in inflammatory gene expression and cytokine secretion (Fig 4f,g). Together, this indicates that loss of *Acly* deregulates fatty acid synthesis and metabolism, where a deregulated PGE₂-EP4 axis might partly explain the altered inflammatory response of *Acly*^{M-KO} macrophages. These metabololipidomics findings are now included in the manuscript and discussed as follows:

*Conversely, levels of omega 3 and 6 fatty acids, including arachidonic acid (AA) tended to be increased in *Acly*^{M-KO} macrophages (Fig. 4c,d). COX-1 and COX-2 produce prostaglandins from AA and these lipid mediators are known to affect inflammatory responses²⁴. The genes encoding COX-1 and COX-2 (*Ptgs1* and *Ptgs2*) and downstream prostaglandins (PGD₂, PGF_{2a}, and in particular PGE₂) were deregulated in *Acly*^{M-KO} macrophages (Fig. 4d). Moreover, *Acly*^{M-KO} macrophages showed a decreased expression of prostaglandin E 4 receptor (EP4, *Ptger4*) (Fig. 4e). Loss of EP4 has previously been described to enhance inflammatory responses^{25,26}. Subsequently, PGE₂- and LPS-treated *Acly*^{M-KO} macrophages display a reduced responsiveness to PGE₂ when compared to control macrophages (Fig. 4f,g).*

*Together, these findings indicate that *Acly*^{M-KO} macrophages rewire cholesterol and fatty acid metabolism to deal with the absence of *Acly*, and these metabolic changes are accompanied by altered inflammatory responses.*

Fig 4. | Deletion of *Acly* leads to deregulation of lipid mediators with decreased responsiveness to prostaglandin E2. (a) Total sum of measured lipids by metabololipidomics. (b) Neutral lipids analyzed by Bodipy 493/503. (c) Sum of AA, DHA, EPA, AdA, LA, ALA/GLA, DPAn-3, and DGLA omega 3 and 6 fatty acids measured by metabololipidomics. (d) Integrative figure of lipid abundances and gene regulation involved in prostaglandin synthesis. (e) *Ptger4* expression in control and *Acly* deficient macrophages (f) Effect of PGE₂ treatment on cytokine secretion in control and *Acly*^{M-KO} LPS-treated BMDMs. (g) effect of PGE₂ treatment on cytokine secretion in control and *Acly*^{M-KO} LPS-treated BMDMs. Values represent mean ± SEM (n=3) with *p < 0.05, **p < 0.01 by two-tailed Student's *t*-test

2. Knockout of *Acly*, theoretically, will decrease or block cholesterol and free fatty acid biosynthesis, which will induce cell death and make the atherosclerosis worse.

Theoretically, one would indeed expect reduced cholesterol and fatty acid synthesis in the absence of *Acly*. However, the macrophages rewire their metabolism in response to knockdown of *Acly* and secure cholesterol and fatty acid synthesis. Nevertheless, *Acly*-deficient macrophages do show a slight increase in apoptosis but this is accompanied by more efficient clearance of apoptotic cells (i.e. efferocytosis) and as such does not worsen atherosclerosis (See Figure 5 in the manuscript).

3. Inhibitors of acetyl CoA carboxylase (ACC) have been used in clinical trials. The enzyme is different from *Acly*. ACC is the first key enzyme for biosynthesis of cholesterol and free fat acids. *Acly* is to provide acetyl CoA but acetyl CoA can also be provided by pyruvate oxidation from glycolysis, which is the major source of acetyl CoA. Therefore, its mechanism has a big question. Histone acetylation data will help a lot.

Pyruvate oxidation via the pyruvate dehydrogenase complex (PDC) is indeed the major source of acetyl-CoA in mitochondria, whereas *Acly* provides cytosolic acetyl-CoA for synthesis of cholesterol and fatty acids downstream of ACC.

As also detailed in response to the first question of this reviewer, decreased histone acetylation is indeed a possible mechanism for decreased IL-4-induced macrophage activation in the absence of *Acly*. These new data are now included in Supplementary Figure 2f and confirm the earlier observations by Covarrubias et al¹⁸.

Conversely, histone acetylation levels in naïve and LPS-treated macrophages are not affected by *Acly*-deficiency. Thus, histone acetylation data do help to explain the phenotype of IL-4-treated macrophages, but not that of naïve and LPS-treated cells. Indeed, in these settings altered cholesterol and fatty acid metabolism upon *Acly* deletion provides mechanistic insight into the resulting macrophage naïve and LPS-induced phenotypes. More specifically, reduced LXR signalling, along with altered levels of lipid mediators that control inflammation, provide an explanation for increased LPS responses in the absence of *Acly*. These mechanisms are described in Figure 3, supplementary figure 4 and Figure 4 and discussed in the corresponding text related to these figures.

Reviewer #3 (Remarks to the Author):

The manuscript by Baardman et al reports some interesting data on the potential role of the enzyme ATP citrate lyase (ACLY) in macrophage phenotype and inflammatory function. Specifically, they show that ACLY is phosphorylated within inflammatory macrophages and in atherosclerotic plaques, and, furthermore, that ACLY deletion in myeloid cells alters the phenotype of these plaques, albeit not macrophage inflammatory function in vitro. Mechanistically, they propose this is due to alterations in macrophage cholesterol metabolism and also to increased apoptosis and efferocytosis of ACLY-deficient macrophages. These data are interesting, novel and potentially of significance for clinical translation and identification of therapeutic targets, however I feel that the data as they are presented don't fully support the conclusions drawn, and additionally that some important mechanistic insight is lacking.

Significant concerns:

1. *The authors genetically manipulate ACLY expression in macrophages, because they show this enzyme to be phosphorylated (and therefore presumably more active) in both inflammatory macrophages in vitro, and also in atherosclerotic plaques in vivo (but see minor concern below). However, they do not confirm increased activity of ACLY in inflammatory macrophages, by either measuring enzyme activity or by analysis of the abundance of the substrate (citrate) or product (acetyl-coA) of this enzyme, which would be important supportive data for their hypothesis. They also do not perform any metabolic analysis of ACLY-deficient macrophages to confirm that their genetic manipulation results in the expected metabolic changes, and assess for any other metabolic implications of ACLY deletion. Indeed, no metabolic analyses are performed in this manuscript, which would provide significant mechanistic insight into how this metabolic enzyme impacts macrophage phenotype and development of atherosclerotic plaques.*

Indeed, western blot analysis (Fig. 1a) showed an increased phosphorylation of ser⁴⁵⁴ in inflammatory macrophages and this is described to increase enzymatic activity²⁰. As suggested by the reviewer, we now also performed in-depth metabolic analyses to define the metabolic implications of Acly deletion:

First, we performed metabolomics analysis to confirm that Acly-deficiency results in the expected increase of its substrate citrate, particularly in response to LPS stimulation (Supplementary Fig. 3b). Other TCA cycle intermediates were not affected by Acly deletion and acetyl-CoA was mostly below the detection limit. However, if detectable, its levels appeared rather increased in the absence of Acly. This could be explained by a compensatory increase in Acss2 (converting acetate in acetyl-CoA) and/or by the fact that pyruvate oxidation by mitochondrial pyruvate dehydrogenase complex PDC is the major contributor to the total acetyl-CoA pool within the cell.

Next, we performed distinct extracellular flux (Seahorse) analyses. Seahorse analysis on naïve macrophages revealed no differences in glycolysis or mitochondrial oxygen consumption between control and Acly^{M-KO} macrophages. Moreover, the expected LPS-induced metabolic switch was similar in control and Acly-deficient macrophages, as shown by comparable glycolysis parameters, OXPHOS parameters and ATP production rates via both pathways.

In line with the Seahorse analysis, glucose usage and lactate production were identical in control and Acly^{M-KO} macrophages as measured in supernatants and also total ATP levels appeared similar.

Together, these data reveal no major defects in the assessed core metabolic pathways and this notion is supported by the similar energy charge calculated based on AMP, ADP and ATP levels in control and KO cells. Mitochondrial analysis by seahorse, glycolysis parameters, cellular ATP concentrations and energy charge data is now indicated in Supplementary Figure 3b-g.

Since our RNA-sequencing data revealed that fatty acid and cholesterol metabolism are altered upon Acly deficiency in macrophages, we further focussed on these pathways as detailed in response to the next questions.

Metabolic analysis are discussed in the manuscript as follows:

To identify the mechanism by which myeloid deletion of Acly influences macrophage and plaque phenotype, we explored changes in core metabolic pathways as a potential explanation of the increased inflammatory response in Acly-deficient macrophages.

Metabolomics revealed that the expected LPS-induced accumulation of citrate was more pronounced in the absence of Acly, signifying decreased Acly-mediated conversion of citrate into acetyl-CoA^{19,20}. Yet, loss of Acly led to no further changes in the abundance of other TCA cycle intermediates (supplementary Fig. 3b). Extracellular flux analysis, and glucose, lactate and ATP measurements showed no differences in mitochondrial function, glycolysis and ATP production, indicating that other mechanisms should explain the observed inflammatory phenotypes of Acly-deficient macrophages (supplementary Fig. 3c-g).

Supplementary Fig 3. Mitochondrial function remains unaltered in *Acly*^{M-KO} BMDMs (a) Differentiated control and *Acly*^{M-KO} macrophages were left untreated or stimulated with LPS (b) relative abundance of TCA cycle intermediates (c) Lactate secretion after 24 hours LPS treatment (d) Metabolized glucose in 24 hours (e) Cellular energy charge as calculated by $[ATP]+0.5[ADP]/([AMP]+[ADP]+[ATP])$ (f) Cellular ATP concentration (g) Seahorse analyses and extracted parameters. Values represent mean \pm SEM (n=6 technical replicates from 3 pooled mice) ***P < 0.001 by two-way ANOVA with Bonferroni post hoc test for multiple comparisons

2. *The authors have performed in-depth transcriptional analyses (RNA-seq) of WT and ACLY-deficient macrophages. These data highlight some interesting differences between the two cell types, for example in cholesterol and fatty acid metabolism, however without any accompanying metabolic analysis it is challenging to mechanistically link ACLY deficiency with the transcriptional changes observed. For example, the authors speculate that the changes in cholesterol metabolism are caused by reduced availability of acetyl-coA (as a compensatory mechanism), but they have not confirmed this.*

Even if (as expected) acetyl-CoA is reduced in abundance, to formally link this to altered cholesterol metabolism, additional experiments using an “add-back” approach or an alternative, pharmacological approach to modulate acetyl-CoA abundance would be very helpful.

As stated by the reviewer, one would expect reduced acetyl-CoA abundance in the absence of *Acly*. While acetyl-CoA was below the detection limit in most of our experiments, in the situations where we were able to detect it, acetyl-CoA levels appeared rather increased in the absence of *Acly*. Since these levels are at too close to the detection limit, these data are not fully trustworthy and as such we prefer to only show them here in response to the reviewer:

Fig Reb. 3 Acetyl-CoA levels in control and *Acly* deficient macrophages as measured by LC-MS. Levels were often below or only just above detection limit and therefore should be interpreted with caution.

Since these data could be explained by a compensatory increase in *Acss2* (the enzyme converting acetate into acetyl-CoA)^{21,22}, we also measured the expression of this gene. Indeed, we observed increased *Acss2* expression in *Acly*^{M-KO} macrophages, which is in line with earlier reports that describe induction of *Acss2* in adipocytes and mouse embryonic fibroblasts upon *Acly* deletion^{21,23}. Together, some compensation mechanisms secured cholesterol and fatty acid synthesis and resulted in normal histone acetylation levels in naïve and LPS-treated control and *Acly*-deficient macrophages (detailed in response to question X of reviewer X and Supplementary Figure 2f).

As an “add-back” approach suggested by the reviewer, we treated macrophages with acetate, followed by LPS stimulation. Interestingly, acetate pre-treatment increased inflammatory responses in both control and *Acly* depleted macrophages and this confirms earlier studies describing the pro-inflammatory effects of the acetate->*Acss2*->acetyl-CoA path^{22,24}. Moreover, pharmacological inhibition of *Acss2* resulted in decreased LPS-induced expression of *Nos2* as a typical inflammatory gene, but increased *Ilf6* expression and had no effect on *Tnf*. Together, these new data indicate the importance of the acetate-*Acss2*-acetyl-CoA axis in *Acly* deficient macrophages but also that this compensatory mechanism alone does not explain the observed macrophage phenotype. This aligns with our data that rewired cholesterol and fatty acid metabolism affects the inflammatory state of macrophages.

Fig Reb. 4 (a) Relative expression of LPS-induced *Il6*, *Nos2*, and *Tnf* with and without acetate in control and *Acly*^{M-KO} macrophages. (b) Expression of *Acss2* in control and *Acly*^{M-KO} macrophages (c) Relative change of *Il6*, *Nos2*, and *Tnf* expression of *Acss2* inhibitor treatment in LPS-activated control and *Acly*^{M-KO} macrophages.

3. The authors also speculate that decreased LXR activity, secondary to reduced abundance of the LXR-activator desmosterol, could explain the elevated inflammatory responses and increased apoptosis of *ACLY*-deficient macrophages. Again, this could be formally tested by adding back desmosterol and assessing inflammatory macrophage phenotype and function, as well as cell cycle progression and apoptosis, but this experiment and the potential key mechanistic insight it could provide is lacking.

To formally test if reduced LXR activity is responsible for the observed increase in inflammatory responses, we used the LXR-agonists GW3965 in control and *Acly*^{M-KO} macrophages. Supporting the presumed anti-inflammatory role of LXR, treatment with LXR-agonist GW3965 decreased inflammatory responses in both control and *Acly*^{M-KO} macrophages. This indicates that LXR activation does indeed reduce inflammatory response as we hypothesized, but this pathway alone does not provide the complete mechanism. Clearly, the effect of *Acly* deletion on the macrophage phenotype is multifactorial and includes a compensation by *Acss2* induction, altered cholesterol metabolism and particularly LXR activity, and altered fatty acid metabolism and associated levels of lipid mediators that control inflammation (as detailed in the answer to question 2 of this reviewer and question 1 of reviewer 2)

These new data are inserted into the manuscript as follows:

Indeed, LXR activation is involved in reprogramming fatty acid metabolism and inhibiting LPS-induced inflammatory-response genes in macrophages²¹. Supporting this

*hypothesis, exposing macrophages to LXR agonist GW3965 decreased inflammatory responses in both control and *Acly*^{M-KO} macrophages (Supplementary Fig. 4a).*

Supplementary Fig 4. GW3965 affects control and *Acly*^{M-KO} macrophages (a) Inflammatory gene expression after GW3965 treatment relative to untreated LPS stimulated gene expression (=100%). Values represent mean \pm SEM (n=4 technical replicates from 3 pooled mice) *P < 0.05 by t-test

4. *In figure 1l the authors show that *Tgfb1* transcripts are increased in the aortic arch of *ACLY-MKO* mice, with the inference that this cytokine is having a pro-fibrotic, anti-atherogenic role. To prove this, it would be important to measure not only mRNA but also (active) TGF- β protein and to formally test its role by neutralisation/signalling blockade in vivo, using specific antibodies or pharmacological inhibitors.*

We speculated in our manuscript about the role of increased *Tgfb1* expression in the aortic arches. TGF- β is known as a pro-fibrotic cytokine in the plaque microenvironment which is upregulated after efferocytosis^{25,26}. Since we saw increased cap thickness, increased efferocytosis and an increased expression of the *Tgfb* gene in the aortic arches we linked these observations. However, we did indeed not test TGF- β protein. Accordingly, we stained for TGF- β protein in the aortic roots²⁷. Co-staining of TGF- β and the macrophage marker MOMA-2 suggested an increase of macrophage TGF- β but not total TGF- β . Since the increased *Tgfb1* levels were subtle, it was probably caused by macrophages within the lesion.

These new data are now shown in Figure 1q-s in the manuscript and described as follows:

*Since labeling of TGF- β and the macrophage marker MOMA-2 suggested an increase of macrophage TGF- β but not total TGF- β (Fig. 1q-s), the increased *Tgfb1* levels were probably caused by macrophages within the lesion.*

Fig 1. (q) Immunohistochemical analysis of macrophages (MOMA-2) and TGF- β in mouse plaques from control or *Acly*^{M-KO} mice. Scale bar represents 100 μ m. **(r)** Quantification of co-localization, percentage of TGF- β ⁺ area in MOMA-2⁺ area. **(s)** Quantification of TGF- β ⁺ area in the lesion.

Formally testing the role of TGF- β protein in the role of atherogenesis in *LDLr*^{-/-} control and *Acly*^{M-KO} mice is definitely an interesting suggestion by the reviewer. However, this will entail a whole new research question in the field about the role of TGF- β in atherogenesis and is therefore beyond the scope of this manuscript. With regards to our observed control-*Acly*^{M-KO} difference, our data suggest

that TGF- β is not the only factor explaining the phenotype, and therefore we felt it was not appropriate to perform another *in vivo* experiment (especially after lockdown of our facilities).

Minor concerns:

1. ***The legend for Figure 1a doesn't accurately describe the data presented – mRNA data as well as protein data are shown but only the protein/WB data are described.***

Description of mRNA has been included accordingly and is highlighted in the legend of Figure 1a.

2. ***The resolution and scale of the image presented in Figure 1b make it very difficult to discern if p-ACLY and CD68 are co-localised. Can the authors improve this? Also it may be helpful to have some quantitation of co-localisation in the image. Furthermore, total ACLY staining (i.e. phosphorylated and non-phosphorylated) is an important control which is lacking here.***

The resolution and scale of the image has been optimized accordingly. As suggested by the reviewer, we analysed co-localization of p-Acly with CD68. P-Acly showed to be mainly co-localized with CD68⁺ cells. Total p-Acly⁺ area shows a significant increase in unstable plaques. These new data are now presented in Fig 1b-d.

Whereas total Acly was present in most cells in and around the plaque, activated (p-) Acly predominantly co-localized with CD45⁺ CD68⁺ macrophages in human atherosclerotic plaques (Fig. 1b,c, Supplementary figure 1a-c). Additionally, unstable plaques showed increased abundance of activated Acly when compared to stable plaques (Fig. 1d). Rupture-prone plaque areas are known to be dominated by inflammatory macrophages¹³ and together with our observation that the levels of activated p-Acly were increased in inflammatory macrophages, this prompted us to study whether targeting macrophage Acly could improve atherosclerosis outcome.

Fig 1. (b) Immunohistochemical staining for macrophages (CD68) and p-ACLY in human plaques. Scale bar represents 100 μ m. **(c)** Quantification of co-localization, percentage of p-Acly⁺ macrophages overlapping with CD68⁺ area. **(d)** Quantification of p-Acly⁺ area in the lesion.

To verify p-Acly staining, we also stained for total Acly protein in the plaque as suggested by the reviewer. Total Acly staining demonstrates positivity in almost all cells throughout the plaque and surrounding cells. Plaques of control and Acly^{M-KO} mice revealed a similar pattern of total Acly staining as human plaques. Knockdown of Acly is clearly visible in the plaque but remains unaltered in the surrounding tissue. These new figures have been added to supplementary figure 1a and b. All in all, based on the stainings in human and mouse plaques we concluded that Acly is an enzyme present in almost all cells, but mainly becomes activated in macrophages.

Supplementary Fig 1. (c) Immunohistochemical staining for CD68 and total Acly

Supplementary Fig 2. (a) Immunohistochemical staining of total Acly on lesions from control and $Acly^{M-KO} Ldlr^{-/-}$ mice.

3. ***ACLY abundance in WT vs KO is only shown in macrophages. It would be helpful to see the level of knockdown in other cell types, particularly other myeloid cells.***

To analyse Acly knockdown in other cell types as suggested by the reviewer, we isolated neutrophils and cultured dendritic cells (BMDCs) and macrophages (BMDMs) from bone marrow of control and $Acly^{M-KO}$ mice. Both dendritic cells and macrophages showed a 3.5 fold reduction of Acly (Supplementary figure 1e). Neutrophils on the other hand revealed minimal *Acly* gene expression in both control and $Acly^{M-KO}$ mice and are therefore not expected to contribute to the observed plaque phenotypes.

Supplementary Fig 1. (e) Relative gene expression of *Acly* in bone marrow derived macrophages, dendritic cells and neutrophils.

As a resemblance of the *in vivo* situation, we now analyzed the expression of *Acly* in peritoneal macrophages from control and *Acly*^{M-KO} *Ldlr*^{-/-} mice. Usually, a peritoneal lavage of *Ldlr*^{-/-} yields a large fraction of foamy macrophages, and therefore it is reflecting the lipid laden macrophage as seen in the plaque¹⁰. In line with *Acly* knockdown in other myeloid cells, peritoneal macrophages obtained from *Ldlr*^{-/-} mice reveal a 3.5 fold reduction of *Acly* gene expression after bone marrow transplantation from control and *Acly*^{M-KO} mice. These new data are now added to the manuscript in Supplementary figures 1e and 2e.

Supplementary Fig 2. (e) *Acly* gene expression in untreated or 24 hour LPS treated peritoneal cells from control and *Acly*^{M-KO} *LDLr*^{-/-} mice.

4. ***In the legend of figure 2 the statement “Representative histograms and quantified surface expression ($\Delta MFI = [Median\ fluorescence\ intensity]_{positive\ staining} - [MFI]_{control}$) are shown” is given but it is unclear whether the “control” is the WT or the paired WT or KO – please can the authors clarify this. It would be most appropriate to use the paired WT or KO respectively.***

In order to control for background signals and false positives, we subtracted the median fluorescence intensity of the indicated marker by an isotype control or a Fluorescence Minus One (FMO) control. The control consists of a mixed phenotype of both control and *Acly*^{M-KO} samples, to prevent measuring differences in autofluorescence between the two cell types. The figure legend is updated for clarification as highlighted and mentioned in the method section as follows:

MFI Controls (isotype or FMO) were measured on a pool of all samples.

References

- 1 Allahverdian, S., Chehroudi, A. C., McManus, B. M., Abraham, T. & Francis, G. A. Contribution of intimal smooth muscle cells to cholesterol accumulation and macrophage-like cells in human atherosclerosis. *Circulation* **129**, 1551-1559, doi:10.1161/CIRCULATIONAHA.113.005015 (2014).
- 2 Bennett, M. R., Sinha, S. & Owens, G. K. Vascular Smooth Muscle Cells in Atherosclerosis. *Circ Res* **118**, 692-702, doi:10.1161/CIRCRESAHA.115.306361 (2016).
- 3 Chappell, J. *et al.* Extensive Proliferation of a Subset of Differentiated, yet Plastic, Medial Vascular Smooth Muscle Cells Contributes to Neointimal Formation in Mouse Injury and Atherosclerosis Models. *Circ Res* **119**, 1313-1323, doi:10.1161/CIRCRESAHA.116.309799 (2016).
- 4 Rong, J. X., Shapiro, M., Trogan, E. & Fisher, E. A. Transdifferentiation of mouse aortic smooth muscle cells to a macrophage-like state after cholesterol loading. *Proceedings of the National Academy of Sciences* **100**, 13531-13536 (2003).
- 5 Thorp, E., Cui, D., Schrijvers, D. M., Kuriakose, G. & Tabas, I. Mertk receptor mutation reduces efferocytosis efficiency and promotes apoptotic cell accumulation and plaque necrosis in atherosclerotic lesions of apoE^{-/-} mice. *Arterioscler Thromb Vasc Biol* **28**, 1421-1428, doi:10.1161/ATVBAHA.108.167197 (2008).
- 6 Kim, K. *et al.* Transcriptome Analysis Reveals Nonfoamy Rather Than Foamy Plaque Macrophages Are Proinflammatory in Atherosclerotic Murine Models. *Circ Res* **123**, 1127-1142, doi:10.1161/CIRCRESAHA.118.312804 (2018).
- 7 Fernandez, D. M. *et al.* Single-cell immune landscape of human atherosclerotic plaques. *Nat Med* **25**, 1576-1588, doi:10.1038/s41591-019-0590-4 (2019).
- 8 Spann, N. J. *et al.* Regulated accumulation of desmosterol integrates macrophage lipid metabolism and inflammatory responses. *Cell* **151**, 138-152, doi:10.1016/j.cell.2012.06.054 (2012).
- 9 Hoeksema, M. A. *et al.* Targeting macrophage Histone deacetylase 3 stabilizes atherosclerotic lesions. *EMBO Mol Med* **6**, 1124-1132, doi:10.15252/emmm.201404170 (2014).
- 10 Baardman, J. *et al.* A Defective Pentose Phosphate Pathway Reduces Inflammatory Macrophage Responses during Hypercholesterolemia. *Cell Rep* **25**, 2044-2052 e2045, doi:10.1016/j.celrep.2018.10.092 (2018).
- 11 Neele, A. E. *et al.* Myeloid Kdm6b deficiency results in advanced atherosclerosis. *Atherosclerosis* **275**, 156-165, doi:10.1016/j.atherosclerosis.2018.05.052 (2018).
- 12 VanderLaan, P. A., Reardon, C. A., Thisted, R. A. & Getz, G. S. VLDL best predicts aortic root atherosclerosis in LDL receptor deficient mice. *J Lipid Res* **50**, 376-385, doi:10.1194/jlr.M800284-JLR200 (2009).
- 13 Lede, V. *et al.* Severe Atherosclerosis and Hypercholesterolemia in Mice Lacking Both the Melanocortin Type 4 Receptor and Low Density Lipoprotein Receptor. *PLoS One* **11**, e0167888, doi:10.1371/journal.pone.0167888 (2016).
- 14 Veniant, M. M. *et al.* Susceptibility to atherosclerosis in mice expressing exclusively apolipoprotein B48 or apolipoprotein B100. *J Clin Invest* **100**, 180-188, doi:10.1172/JCI119511 (1997).
- 15 Groot, P. H. *et al.* Quantitative assessment of aortic atherosclerosis in APOE* 3 Leiden transgenic mice and its relationship to serum cholesterol exposure. *Arteriosclerosis, thrombosis, and vascular biology* **16**, 926-933 (1996).
- 16 Lauterbach, M. A. *et al.* Toll-like Receptor Signaling Rewires Macrophage Metabolism and Promotes Histone Acetylation via ATP-Citrate Lyase. *Immunity* **51**, 997-1011 e1017, doi:10.1016/j.immuni.2019.11.009 (2019).

- 17 Langston, P. K. *et al.* Glycerol phosphate shuttle enzyme GPD2 regulates macrophage inflammatory responses. *Nat Immunol* **20**, 1186-1195, doi:10.1038/s41590-019-0453-7 (2019).
- 18 Covarrubias, A. J. *et al.* Akt-mTORC1 signaling regulates Acly to integrate metabolic input to control of macrophage activation. *Elife* **5**, doi:10.7554/eLife.11612 (2016).
- 19 Namgaladze, D. *et al.* Polarization of Human Macrophages by Interleukin-4 Does Not Require ATP-Citrate Lyase. *Front Immunol* **9**, 2858, doi:10.3389/fimmu.2018.02858 (2018).
- 20 Potapova, I. A., El-Maghrabi, M. R., Doronin, S. V. & Benjamin, W. B. Phosphorylation of recombinant human ATP:citrate lyase by cAMP-dependent protein kinase abolishes homotropic allosteric regulation of the enzyme by citrate and increases the enzyme activity. Allosteric activation of ATP:citrate lyase by phosphorylated sugars. *Biochemistry* **39**, 1169-1179, doi:10.1021/bi992159y (2000).
- 21 Liu, X. *et al.* Acetate Production from Glucose and Coupling to Mitochondrial Metabolism in Mammals. *Cell* **175**, 502-513 e513, doi:10.1016/j.cell.2018.08.040 (2018).
- 22 Leone, R. D. *et al.* Glutamine blockade induces divergent metabolic programs to overcome tumor immune evasion. *Science* **366**, 1013-1021 (2019).
- 23 Zhao, S. *et al.* ATP-Citrate Lyase Controls a Glucose-to-Acetate Metabolic Switch. *Cell Rep* **17**, 1037-1052, doi:10.1016/j.celrep.2016.09.069 (2016).
- 24 Kendrick, S. F. *et al.* Acetate, the key modulator of inflammatory responses in acute alcoholic hepatitis. *Hepatology* **51**, 1988-1997, doi:10.1002/hep.23572 (2010).
- 25 Thomas, A. C., Eijgelaar, W. J., Daemen, M. J. & Newby, A. C. Foam Cell Formation In Vivo Converts Macrophages to a Pro-Fibrotic Phenotype. *PLoS One* **10**, e0128163, doi:10.1371/journal.pone.0128163 (2015).
- 26 Hoeksema, M. A. *et al.* Targeting macrophage Histone deacetylase 3 stabilizes atherosclerotic lesions. *EMBO molecular medicine* **6**, 1124-1132 (2014).
- 27 Cipollone, F. *et al.* Increased expression of transforming growth factor-beta1 as a stabilizing factor in human atherosclerotic plaques. *Stroke* **35**, 2253-2257, doi:10.1161/01.STR.0000140739.45472.9c (2004).

REVIEWERS' COMMENTS

Reviewer #1 (Remarks to the Author):

The authors have performed a large number of new experiments to address the reviewers' comments and questions, and the manuscript is improved as a result. However, I don't agree with the conclusion that the plaque phenotype is in part due to increased efferocytosis, as implied in the abstract and elsewhere. The new in vivo data suggest that efferocytosis was similar, but not reduced. The conclusion "This results in macrophages that are more prone to undergo apoptosis, whilst maintaining their capacity to phagocytose apoptotic cells" or something similar would better reflect the data.

Reviewer #2 (Remarks to the Author):

The authors have answered all my comments. I have no further suggestion.

Reviewer #3 (Remarks to the Author):

The authors have now revised their manuscript in response to the reviewers' comments, including substantial additional experimental work and re-analysis of data.

I feel they have satisfied the majority of my comments in doing so. Certainly all of the minor comments have been satisfactorily addressed, aiding the clarity of the figures and legends for readers. I think it is also really helpful to see the total ACLY staining and indeed this strengthens the observation that phospho-ACLY is selectively enriched in CD68+ cells.

The TGFb protein labeling (major comment 4) is a little less convincing - specifically that this protein is particularly enriched within macrophages in the context of the ACLY knockdown. It doesn't appear that this trend would be statistically significant. However, I appreciate that TGFb protein staining is technically challenging and it is also not a key finding of the paper upon which other findings depend, therefore I am satisfied with the response provided.

In response to the more significant comments, the authors have now undertaken in-depth metabolic analyses including measuring the abundance of TCA cycle intermediates and performing metabolic flux analyses of control and ACLY-deficient macrophages. I think this is really helpful to aid interpretation of their findings and certainly addresses what felt like a key gap between manipulating expression of a metabolic gene and observing phenotyping and functional differences in vitro and in vivo.

These new data reveal that the mechanistic basis underlying the altered phenotype of ACLY-deficient macrophages is perhaps more complicated than originally proposed, since central metabolic changes are surprisingly few (apart from expected changes in citrate abundance). However, I think the authors handle this well, providing alternative explanations from their sequencing data, which are also interrogated (e.g. the new LXR agonist experiment). One question remaining is regarding ACSS2. It seems that the increase in ACSS2 expression in the absence of ACLY is quite significant and may help to interpret the metabolic findings. I therefore wonder whether this should be included in a main or supplementary figure and discussed in the text? Conversely, whilst I appreciate that they undertook the acetate "add-back" experiments (and I found these very interesting and helpful) I agree with the authors that their results may further complicate the interpretation of the manuscript since acetate clearly has a pro-inflammatory effect, even in control macrophages and in any case acetyl-coA is not diminished in ACLY-deficient cells, which would be the rationale for this approach.

We would like to thank the reviewers for their careful review of our manuscript and their help to improve the interpretation of our findings. We are pleased to hear that the reviewers are satisfied with the additional work we did, and are happy to publish a suitable revised version of our manuscript.

We now resubmit the revised version of our paper where we included the comments suggested by reviewer 1 and 3.

Reviewer #1 (Remarks to the Author):

The authors have performed a large number of new experiments to address the reviewers' comments and questions, and the manuscript is improved as a result. However, I don't agree with the conclusion that the plaque phenotype is in part due to increased efferocytosis, as implied in the abstract and elsewhere. The new in vivo data suggest that efferocytosis was similar, but not reduced. The conclusion "This results in macrophages that are more prone to undergo apoptosis, whilst maintaining their capacity to phagocytose apoptotic cells" or something similar would better reflect the data.

In the revised version of our manuscript, we addressed the point raised by the reviewer by rephrasing our statement in the abstract as suggested by the reviewer: 'This results in macrophages that are more prone to undergo apoptosis, whilst maintaining their capacity to phagocytose apoptotic cells.'

Reviewer #2 (Remarks to the Author):

The authors have answered all my comments. I have no further suggestion.

We would like to thank the reviewer for their comments and remarks on our earlier version of the manuscript to help to improve.

Reviewer #3 (Remarks to the Author):

The authors have now revised their manuscript in response to the reviewers' comments, including substantial additional experimental work and re-analysis of data.

I feel they have satisfied the majority of my comments in doing so. Certainly all of the minor comments have been satisfactorily addressed, aiding the clarity of the figures and legends for readers. I think it is also really helpful to see the total ACLY staining and indeed this strengthens the observation that phospho-ACLY is selectively enriched in CD68+ cells.

The TGFb protein labeling (major comment 4) is a little less convincing - specifically that this protein is particularly enriched within macrophages in the context of the ACLY knockdown. It doesn't appear that this trend would be statistically significant. However, I appreciate that TGFb

protein staining is technically challenging and it is also not a key finding of the paper upon which other findings depend, therefore I am satisfied with the response provided.

In response to the more significant comments, the authors have now undertaken in-depth metabolic analyses including measuring the abundance of TCA cycle intermediates and performing metabolic flux analyses of control and ACLY-deficient macrophages. I think this is really helpful to aid interpretation of their findings and certainly addresses what felt like a key gap between manipulating expression of a metabolic gene and observing phenotyping and functional differences in vitro and in vivo.

These new data reveal that the mechanistic basis underlying the altered phenotype of ACLY-deficient macrophages is perhaps more complicated than originally proposed, since central metabolic changes are surprisingly few (apart from expected changes in citrate abundance). However, I think the authors handle this well, providing alternative explanations from their sequencing data, which are also interrogated (e.g. the new LXR agonist experiment). One question remaining is regarding ACSS2. It seems that the increase in ACSS2 expression in the absence of ACLY is quite significant and may help to interpret the metabolic findings. I therefore wonder whether this should be included in a main or supplementary figure and discussed in the text? Conversely, whilst I appreciate that they undertook the acetate "add-back" experiments (and I found these very interesting and helpful) I agree with the authors that their results may further complicate the interpretation of the manuscript since acetate clearly has a pro-inflammatory effect, even in control macrophages and in any case acetyl-coA is not diminished in ACLY-deficient cells, which would be the rationale for this approach.

In the revised version of the manuscript we now added the figure showing *Acss2* expression in *Acly* deficient macrophages in Figure 6h as suggested by the reviewer. We discussed this rationale as follows:

‘Moreover, *Acly*-deficient macrophages might rescue acetyl-CoA production by upregulating *Acss2* (Fig. 6h). This gene encodes acyl-coenzyme A synthetase short-chain family member 2, an enzyme that converts acetate into acetyl-CoA. This is in line with earlier reports that describe induction of *Acss2* in adipocytes and mouse embryonic fibroblasts upon *Acly* deletion.’ (+reference 20, 26 from the manuscript).

We agree with the reviewer that the fact that acetate has such pro-inflammatory effects, it complicates the interpretation of the findings in the manuscript. Though, by including the data on *Acss2* expression we hope to aid the reader in understanding the manuscript.